# Assessment of ocean analysis and forecast from an atmosphere-ocean coupled data assimilation operational system

Catherine Guiavarc'h[1], Jonah Roberts-Jones[1], Chris Harris[1], Daniel J. Lea[1], Andrew Ryan[1], and Isabella Ascione[1]

[1]Met Office, FitzRoy Road, EX1 3PB, Exeter, United Kingdom

**Correspondence:** Chris Harris (christopher.harris@metoffice.gov.uk)

**Abstract.** The development of coupled atmosphere-ocean prediction systems with utility on the short-range Numerical Weather Prediction (NWP) and ocean forecasting timescales has accelerated over the last decade. This builds on a body of evidence showing the benefit, particularly for weather forecasting, of more correctly representing the feedbacks between surface ocean and atmosphere. It prepares the way for more unified prediction systems with the capability of providing consistent surface me-

teorology, wave and surface ocean products to users for whom this is important. Here we describe a coupled ocean-atmosphere system, with weakly coupled data assimilation, which was operationalised at the Met Office as part of the Copernicus Marine Environment Service (CMEMS). We compare the ocean performance to that of an equivalent ocean-only system run at the Met Office, and other CMEMS products. Sea surface temperatures in particular are shown to verify better than in the ocean-only systems, although other aspects including temperature profiles and surface currents are slightly degraded. We then discuss the

plans to improve the current system in future as part of the development of a "coupled NWP" system at the Met Office.

*Copyright statement.* TEXT

## 1   Introduction

Coupled systems are used in wide range of applications (short and medium range forecasts, seasonal forecasts, climate prediction and future scenario projections) and improving the initialization of these systems can play a significant part to reduce

the development of errors. Using separate atmosphere and ocean analyses to initialize a coupled system can result in an imbalanced system. The imbalance can cause an initialisation shock which potentially increases the development of errors during the forecast. Using a coupled data assimilation (DA) approach has been shown to reduce this initialisation shock (Mulholland et al., 2015).

   For many years, the Met Office has been running short-range prediction systems separately for the atmosphere and the ocean.

Although seasonal forecasts use coupled models, they are initialised from separate uncoupled analyses. Since October 2016, the Met Office coupled atmosphere-land-ocean–ice data assimilation system (CPLDA) has been running operationally and its ocean forecast and analysis have been delivered daily to the Copernicus Marine Environment Monitoring Service (CMEMS)

since July 2017. The CPLDA system is the first operational Met Office system providing a seamless coupled analysis and forecast. It follows work done in the Met Office in recent years to highlight the impact of coupling for the short to medium range forecast (Johns et al., 2012) and to develop a coupled DA system (Lea et al., 2013) that could in the future replace the uncoupled atmosphere and ocean short range prediction systems. In Johns et al. (2012), a large set of global coupled atmosphere-ocean-

sea ice 15-day hindcasts were initialised from separate atmosphere and ocean-ice analyses. Compared to uncoupled forecasts, the coupled 15-day hindcasts show improved forecasting skill especially in the Tropics, a strongly coupled region where small changes in sea surface temperature (SST) can exert a major influence on patterns of convection and remote responses to diabatic heating. Both atmosphere and ocean biases benchmarked against control simulations suggest a more vigorous water cycle, improved large-scale circulation, enhanced convection, stronger teleconnections and somewhat improved representation

of the Madden-Julian oscillations. They conclude that significant benefits should arise from full integration of a coupled NWP system.

In recent years, several centres have developed operational coupled data assimilation systems. Penny and Hamill (2017) provide an overview of many of these efforts, showing the diversity of approaches available. For example, the European Centre for Medium-Range Weather Forecast (ECMWF) has developed a weakly coupled ocean-atmosphere data assimilation sys-

tem (Browne et al., 2019) which is now operational. This system is shown to reduce forecast errors compared with forecasts initialised from uncoupled analyses. The National Centers for Environmental Prediction (NCEP) is using a coupled data assimilation system for seasonal and sub-seasonal predictions (Saha et al., 2014). At JAMSTEC, a low resolution strongly coupled system was developed to be used for experimental seasonal and decadal prediction, while the US Naval Research Laboratory coupled model is initialised from separate analyses but uses a high-resolution ocean component $(1/25°)$. Similarly, the Cana-

dian Centre for Meteorological and Environmental Prediction operational global coupled system uses uncoupled analyses but taking great care to maintain consistency of surface boundary conditions between components in order to avoid initialisation shocks. Smith et al. (2018) show good forecast performance compared to the atmosphere-only counterpart of this system, with the most significant impact associated with decreased tropical cyclone intensification.

The approach taken to develop the CPLDA system is described in detail in Lea et al. (2013, 2015). It is based on the existing

coupled model developed for seasonal forecasting and climate prediction and on existing data assimilation systems for the ocean, sea-ice, land and atmosphere. The data assimilation follows a "weakly coupled" approach. This means the coupled model provides background information for separate analyses in each sub-component with the increments being added back into the coupled model. Lea et al. (2013) assessed the performance of the system against uncoupled control experiments for two short trials (December 2011 and June 2012). In this paper, we present results from the first long simulation run with the

CPLDA system. We run the system for a year (February 2015 to January 2016) with 6-hourly analysis and daily 7-day forecast.

The Met Office Ocean Forecasting R&D group (OFRD) has been providing ocean analyses and forecasts from the Forecast Ocean Assimilation Model (FOAM) for many years. FOAM is an accurate system that verifies well against similar international systems (Ryan et al., 2015) making it a suitable benchmark against which to assess the ocean component of the CPLDA system. In this study, we investigate the added value of an ocean analysis and forecast provided by a coupled system and try to

understand the causes of the differences between the coupled and uncoupled systems. As the primary focus of this study is to

assess the quality of the product delivered to CMEMS, our work focussed mainly on the assessment of the ocean component of the CPLDA system. To avoid duplication of previous work not every aspect has been investigated. For instance, we did not investigate the impact on the diurnal cycle already covered in Lea et al. (2013, 2015). They investigated the impact of the initialisation on the diurnal cycle of SST by comparing to geostationary satellite data from SEVIRI. They looked at the diurnal

SST range and showed that the initialisation of the coupled forecast has very little impact on this. Lea et al. (2015) showed that in the South Pacific the coupled model diurnal cycle errors against observations are somewhat amplified compared to the ocean-only control. However the diurnal cycle is perhaps under-estimated in the central North Atlantic (Lea et al., 2013).

A description of the CPLDA system and of the experimental set-up used to assess the system are presented in section 2, together with a description of the differences with the FOAM configuration. The results are presented in section 3. We focus

on assessment of the ocean analysis and forecast from CPLDA as well as comparison with FOAM analysis and forecast and with the Mercator $1/12°$ analysis (PSY4). Assessment of the differences at the air-sea interface between the coupled system and the uncoupled systems are also presented. Section 4 summarises the main results and provides discussions and plans for future work.

## 2 Description of the coupled data assimilation system and experiments

The weakly coupled atmosphere-land-ocean-ice data assimilation system (CPLDA) is built on the coupled system developed by Lea et al. (2015). Details of the scientific and technical implementation of the system is described in their paper. The CPLDA system has been running operationally since October 2016 and is the first Met Office operational coupled forecasting system with complete consistency between the analyses and the forecast. The coupled forecast is initialised by coupled analysis for both ocean and atmosphere components. This continuity means that the atmosphere and ocean components are identical in the

analysis and in the forecast. The previous coupled forecasting system was using uncoupled analyses from FOAM (Blockley et al., 2014) and NWP (Numerical Weather Prediction) to initialise the GloSea coupled forecast (MacLachlan et al., 2014). In that case both the atmosphere and ocean components differed between the analysis and the forecast. The different components of the CPLDA system are described below and the differences with the FOAM system used for comparison in the next sections are highlighted.

### 2.1 Atmosphere component and surface forcing

In the CPLDA system, analysis and forecast fluxes are calculated using bulk formulae based on COARE3.0 (Fairall et al., 2003) within the Unified Model (MetUM) atmosphere at  40 km resolution and interpolated and passed to the NEMO ocean component by the OASIS3 coupler (Valcke, 2006). The FOAM system used interpolated atmospheric fields from the operational Met Office global NWP configuration of the MetUM (at  17 km resolution in 2015) with CORE bulk formulae (Large

and Yeager, 2004) to calculate the turbulent fluxes. The CPLDA atmospheric component is the GA6.0 (Walters et al., 2017) atmospheric science configuration (also used by GloSea) for both the analysis and forecast while the NWP configuration used to force FOAM was GA6.1 (note that this distinction mainly relates to aspects of the land-surface treatment and is largely

irrelevant from the ocean point of view). The CPLDA atmosphere data assimilation system is described in detail in Lea et al. (2015). It uses an incremental strong constraint 4DVAR system similar to Rawlins et al. (2007). One addition to the system described in Lea et al. (2015) is that the CPLDA atmosphere data assimilation now uses a variational bias correction (VarBC) to continuously update the bias correction applied to observations. VarBC (Lorenc, 2013) was not implemented operationally

in the NWP system until March 2016; however, all the CPLDA experiments in this paper use VarBC.

## 2.2   Ocean component

The ocean configuration is described in detail below and the CPLDA and FOAM ocean components are summarised in Table 1. Both CPLDA and FOAM systems use the global ocean configuration GO5 (Megann et al., 2014). GO5 uses version 3.4 of the NEMO modelling system (Madec and the NEMO team, 2008) with the ORCA025 tripolar horizontal grid (which has a $1/4°$

or 28 km horizontal grid spacing at the equator reducing to 7 km at high southern latitudes, and  10 km in the Arctic Ocean) and is based on the configuration developed by Mercator Ocean. The vertical coordinate system is based on geopotential levels using the DRAKKAR 75 level set which provides an increased near surface resolution (including 1 m surface layers to help resolve shallow mixed layers and potentially capture diurnal variability) without compromising resolution at depth. The model bathymetry is DRAKKAR v3.3 which is based on the ETOPO1 data set (Amante and Eakins, 2009) with additional data in

coastal regions from GEBCO (General Bathymetric Chart of the Oceans; IOC, IHO and BODC). Partial cell thicknesses at the ocean floor allow a better representation of ocean topography and in combination with an energy- and enstrophy-conserving momentum advection scheme and a free slip lateral momentum boundary condition improve the mesoscale circulation, and in particular the simulation of western boundary currents. The tracers are advected using a total-variation-diminishing (TVD) scheme (Zalesak, 1979) and a linear filtered free surface is used to remove high frequency gravity waves. Tracer diffusion is

laplacian along isopycnals, and horizontal momentum diffusion is performed using a bilaplacian operator along geopotential levels. The vertical diffusion implemented in the CPLDA system is the Turbulent Kinetic Energy (TKE) scheme of Gaspar et al. (1990) updated to ensure dynamical consistency in the space/time discretisations (Burchard, 2002). The background diffusivity and viscosity includes a double diffusive mixing parametrisation and a parametrisation to attempt to model the mixing effect of Langmuir circulation. In CPLDA, both analysis and the forecast use a Haney retroaction to control sea surface salinities, a 3D

Newtonian damping towards a World Ocean Atlas 2001 climatology (to prevent long term drift of sub-surface tracer fields) and a pressure gradient bias correction in the Tropics (Bell et al., 2004) to ensure temperature and salinity increments are retained by the model. The Haney flux correction (Haney, 1971) is applied to the sea surface salinity (SSS) based on the difference between the model and climatology; the magnitude of the restoring on SSS is -33.33 mm day$^{-1}$ psu$^{-1}$. A summary of the settings for the ocean component in FOAM and CPLDA is shown in Table  1. Most settings are identical except for necessary

differences in the surface forcing and a shorter assimilation time window for CPLDA. The ocean data assimilation is described in more detail in section 2.5.

## 2.3 Sea ice component

The sea ice model (CICE version 4.1; Hunke and Lipscomb, 2010) runs on the same ORCA025 grid as NEMO, and with five ice thickness categories. The CICE model determines the spatial and temporal evolution of the ice thickness distribution (ITD) due to advection, thermodynamic growth and melt, and mechanical redistribution / ridging (Thorndike et al., 1975). The CPLDA system documented predates the development work described in West et al. (2016) and Ridley et al. (2018). Therefore when running coupled to the Unified Model (MetUM) atmosphere, the CPLDA system uses the zero-layer thermodynamic model of Semtner (1976), with a single layer of both ice and snow in CICE. The FOAM system, on the other hand, uses a five layer thermodynamic model (four ice layers and one snow layer). Ice dynamics are calculated using the elastic-viscous-plastic (EVP) scheme of Hunke and Dukowicz (2002). In CPLDA, for both the analysis and forecast, the ice top and bottom conductive heat fluxes are calculated within the atmosphere model, interpolated by OASIS, and then passed to NEMO from where they can be accessed by CICE. In GloSea, the heat fluxes calculated by the atmosphere model were also used but in the FOAM analysis CICE used its own bulk formulation to specify surface boundary conditions.

In FOAM the freezing temperature is dependent on salinity to provide a more realistic representation of ice melting and freezing mechanisms and to give better consistency when assimilating both sea surface temperature and sea ice concentration. For technical reasons the initial coupled DA system assessed here used a fixed freezing temperature of -1.8°C, as did the GloSea system. However salinity dependence was later introduced in CPLDA in September 2018. The GSI6 global sea-ice configuration used in the CPLDA system is detailed in Rae et al. (2015).

## 2.4 Ocean observations

The ocean observations assimilated into the CPLDA system are the same as in FOAM. These are as follows:

- Satellite SSTs – sub-sampled level 2 data supplied by the Global High-Resolution Sea Surface Temperature (GHRSST) project comprising Advanced Very High Resolution Radiometer data (NOAA & MetOp), microwave AMSR-2 data and Visible Infrared Imaging Radiometer Suite (VIIRS) data. Note that daytime satellite SST data with a strong diurnal signal where the wind speed is less than 6 m s$^{-1}$ is not used.

- In situ SSTs – from moored buoys, drifting buoys and ships (the in-situ observations are considered unbiased and used as a reference for satellite SST bias correction).

- Sea level anomaly (SLA) observations – from Jason-2, Jason-3, CryoSat-2, SARAL/AltiKa and Sentinel-3a platforms.

- Sub-surface temperature and salinity profiles from Argo profiling floats, underwater gliders, moored buoys, marine mammals, and manual profiling methods.

- Sea ice concentration - Special Sensor Microwave Imager/Sounder (SSMIS) data provided by the EUMETSAT Ocean Sea Ice Satellite Application Facility (OSI-SAF) as a daily gridded product on a 10 km polar stereographic projection.

Model fields are mapped into observation space using the NEMO observation operator to create nearest timestep model counterparts at the observation location using bilinear interpolation in the horizontal and cubic splines in the vertical directions.

## 2.5 Ocean data assimilation

The CPLDA system is based on a "weakly coupled" data assimilation approach. The coupled model is used to provide background information for separate ocean, sea-ice, atmosphere and land analyses. The increments generated from these separate analyses are added back into the coupled model (Lea et al., 2015).

For the ocean and sea-ice DA both FOAM and CPLDA use NEMOVAR (Mogensen et al., 2012) – an incremental 3D-Var, first guess at appropriate time (FGAT) assimilation scheme designed specifically for use with NEMO and further tuned at the Met Office for the $1/4°$ global model (Waters et al., 2013, 2015). The state vector in NEMOVAR consists of temperature, salinity, surface elevation, sea ice concentration and horizontal velocities. Key features of NEMOVAR are the multivariate relationships which are specified through a linearised balance operator (Weaver et al., 2006) and the use of an implicit diffusion operator to model background error correlations (Mirouze and Weaver, 2010). As detailed in Waters et al. (2013), the NEMOVAR system includes bias correction schemes for both sea surface temperature (SST) and altimeter data (using the CNES-CLS09 Mean Dynamic Topography (MDT) of Rio et al. (2011) as a reference). The temperature and unbalanced salinity are assimilated using two horizontal correlation length scales (Mirouze et al., 2016) following the method described in Martin et al. (2007). FOAM and CPLDA run one outer loop and 40 iterations of the inner loop minimisation of the cost function. Note that the sea ice DA is run in a separate assimilation step to the ocean DA.

Analysis updates are made to the state variables in the NEMO model, with the exception of sea ice concentration updates which are made in the CICE model. Updates increasing ice concentration are always made to the thinnest category ice, whilst updates decreasing ice concentration are made to the thinnest ice thickness category available in that grid cell (Blockley et al., 2014). The snow thickness on ice is preserved where there is existing ice, but initialised to zero where new ice is added to a previously ice free grid cell (Blockley et al., 2014).

A few technical changes to the FOAM ocean data assimilation system are needed to fit into the coupled framework. The first technical change made was to match the atmosphere cycling. In the FOAM system, the observation operator and incremental analysis update (IAU) are performed in separate NEMO runs. In CPLDA, this has been altered such that the IAU is followed by observation operator in one combined model run. The second major change from FOAM to CPLDA was to reduce the assimilation time window of the ocean from 24 h to 6 h to match the time window of the atmosphere (see Lea et al., 2015). The third change was to reduce the period over which the increment is added to the ocean model in the IAU from 24 h to 3 h. This choice is in part to ensure that on the 00Z cycle (with window from 21Z to 03Z) the full increments have been added to the ocean state by 00Z. Hence these ocean restarts can subsequently be used, in exactly the same way as 00Z ocean restarts from FOAM, by groups running initialised coupled forecasts. It is also important to note that differences in observation cutoff due to difference in operational scheduling impact the number of observations assimilated by each system.

No specific tuning has been made for the coupled system. The error covariances used in CPLDA were calculated in the context of the uncoupled FOAM system with a 24 hour cycle. It would be expected that we would improve the DA results by recalculating the error covariances for a coupled model and for a 6-hour window (this is planned for future work).

## 2.6 Experiments

In order to assess the CPLDA system, we ran a 13-month calibration period from 01/01/2015 to 31/01/2016 with 6-hourly analysis and daily 7-day forecast. To allow parallel running, the 13 months were realised in two sections, each starting from "uncoupled" initial conditions from the FOAM and NWP operational analyses. The first section was from 01/12/2014 to 30/06/2015, whilst the second section was from 01/06/2015 to 31/01/2016. For both sections, the first month is taken as a spin-up period and discarded. No discontinuity has been observed between 06/2015 from the first section and 07/2015 from

the second section and so all assessment undertaken considers the whole of 2015. Every six hours, a delayed "best analysis" runs 24 hours behind real-time with a catch-up to real-time only required to launch the 7-day forecasts on the 00Z cycle. For running of delayed-time trials, like that for 2015 described in this paper, the operational availability of observations is replicated by using receipt time information when they are extracted from the Met Office observations database (MetDB). In the following sections, we assess the CPLDA system against the ocean-only FOAM system for the 2015 period. FOAM

has been used as the Met Office operational ocean forecasting system for many years. It uses atmospheric forcing from the Met Office NWP operational system and is described in detail in Blockley et al. (2014). Differences between the CPLDA and FOAM ocean configurations are shown in Table 1. There are a few differences between the CPLDA system assessed in the majority of this paper and the final operational implementation. As the systems were running for different periods observations not available, or not yet operationally implemented, in 2015 (Jason-3 and Sentinel-3 sea level anomaly data, and VIIRS sea

surface temperatures) are only assimilated in the final operational version. More significantly the scheduling of the operational system has been modified to allow additional observations to be assimilated. The result of these changes is that the "best analysis" runs further behind real time on the 06Z, 12Z and 18Z cyles (42, 36 and 30 hours respectively, compared to 24 hours previously). As for the 2015 trial the catch-up to real time is only required to launch the forecast (operationally this is now 10 days) once per day on the 00Z cycle. Where any results are shown from the CPLDA system running with this modified

scheduling (as opposed to that used in the 2015 trial) this is clearly indicated.

Additional experiments have been undertaken using the ocean-only FOAM system to look at the impact of running the data assimilation with a shorter assimilation window. The FOAM system was run in hindcast mode (forecasts were not generated) for 6 months from 01/03/2018 to 31/08/2018 with the analysis length and assimilation window set to 6 and 24 hours. These experiments should decompose the effect of the shortened assimilation window from the effect of the coupling and aid in the

interpretation of the comparisons to the FOAM system described herein.

## 3  Results

In this section, we present the results from a one year assessment of the CPLDA system from February 2015 to January 2016. A one year experiment is required to obtain representative results and longer experiments are complex to run because of the computational cost and the constant evolution of the observation network, particularly for the atmosphere component. We
assess the CPLDA ocean component against observations and other benchmark operational ocean analyses and forecasts (Met Office FOAM and Mercator $1/12°$ PSY4 (Lellouche et al., 2018)). "Class 4" metrics are widely used to assess the accuracy of ocean forecasting systems. They are statistics of the differences between oceanic observations (in situ or satellite) and their model (forecast or analysis) equivalent at the time and location of the observation (Ryan et al., 2015).

In the following subsections, we present in detail the results from the assessment of the sea surface height, sea ice, sea
surface temperature, three-dimensional temperature, mixed layer depth (MLD) and the currents at 15 m. Although this paper largely presents results from the ocean component of the coupled system, we also discuss the impact of differences in surface forcing between CPLDA and FOAM and how these relate to the atmospheric configurations used in the two systems.

### 3.1  Sea Surface Height

CPLDA 2015 and FOAM sea surface height are assessed against observations using class4 statistics (Ryan et al., 2015).
The observations used are provided by CMEMS and include data from Altika, CryoSat-2 and Jason-2 satellites. Altimeter bias correction is applied to the observations. For each model comparison against the satellite observations, the model's own altimeter bias is used. This is important because the altimeter bias contains information from the model mesoscale so correcting observations using the altimeter bias from one model to assess a second model penalises this second model. Fig 1 shows a timeseries of the sea level anomaly (SLA) difference statistics assessed against CMEMS observations. In the 2015 experiment
with the original scheduling, CPLDA SLA root-mean-squared error (RMSE) is significantly larger than FOAM SLA RMSE (Fig 1a). The larger RMSE in CPLDA can be attributed to the difference in the number of SLA observations assimilated. Fig 2 shows the number of observations assimilated daily by both systems in 2015; the number assimilated by CPLDA is significantly smaller than the number assimilated by FOAM. Differences in scheduling can explain this the CPLDA 2015 experiment "best analysis" runs earlier in the day than the comparable FOAM analysis so fewer observations are available.
Following these results, the scheduling of the CPLDA operational system was updated in April 2018. The best analyses at 06Z, 12Z and 18Z are now delayed allowing more observations to be assimilated, as already described in section 2. This change, along with a change in the Met Office database (MetDB) allowing a more frequent ingestion of SLA observations, has resulted in a significant reduction of the CPLDA RMSE (see Fig 1b). The experiments using the FOAM system to look at the effect of the shorter assimilation window in CPLDA showed no significant difference in SLA statistics between the 6 and 24 hour
systems.

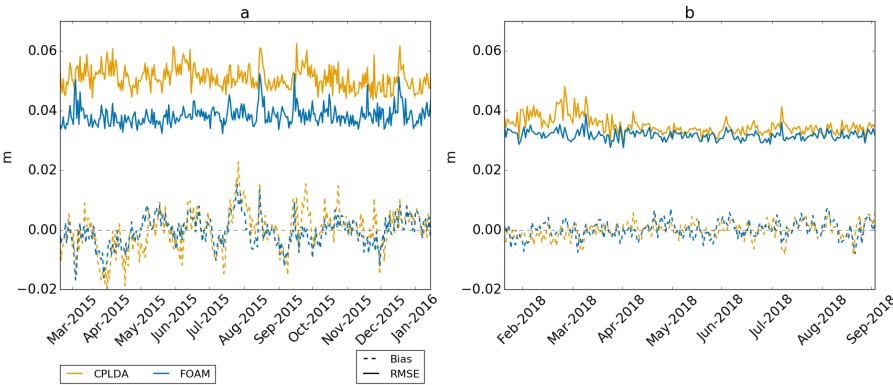

**Figure 1.** SLA class4 statistics with respect to CMEMS satellite product (Jason2, CryoSat-2 and Altika) for CPLDA and FOAM. (a) For the long CPLDA experiment in 2015 with the old scheduling. (b) from the operational systems for 2018, the CPLDA scheduling was updated to allow more observations to be assimilated in April 2018.

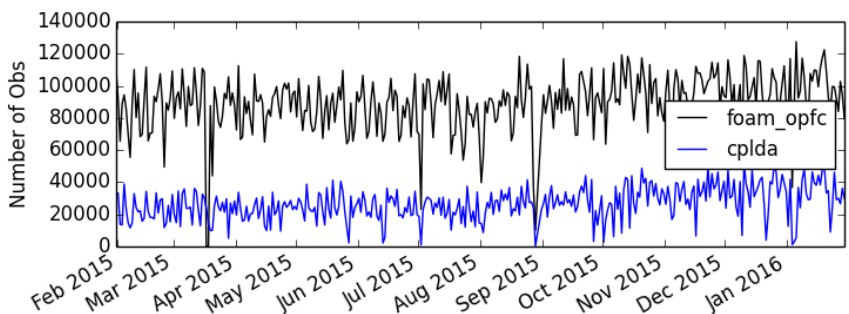

**Figure 2.** Daily number of SLA observations assimilated in FOAM operational system for 2015 (black line, foam opfc) and in the CPLDA 2015 experiment (blue line, cplda)

## 3.2  Sea Ice

Fig 3 shows the sea ice extent and volume respectively for CPLDA and FOAM best analyses. It shows that the sea ice extent is similar between CPLDA and FOAM and is also comparable to OSTIA. Fig 3 shows that the sea ice volumes are similar in the Arctic but in the Antarctic CPLDA has a reduced volume compared to FOAM in the Southern Hemisphere winter and spring. Differences in volume can be attributed to differences in sea ice thickness probably caused by differences in freezing temperature treatment. Fig 4 shows the sea ice extent forecast for the melting season in Arctic (August) and Antarctic (February) for CPLDA and FOAM. Both models have the tendency to melt too much ice during the forecast especially in the Arctic; in the Antarctic the exaggerated melting in CPLDA is slightly reduced compared to that observed in FOAM. Differences in freezing temperature treatment referred to in 2.3 may have impacted the sea ice simulation.

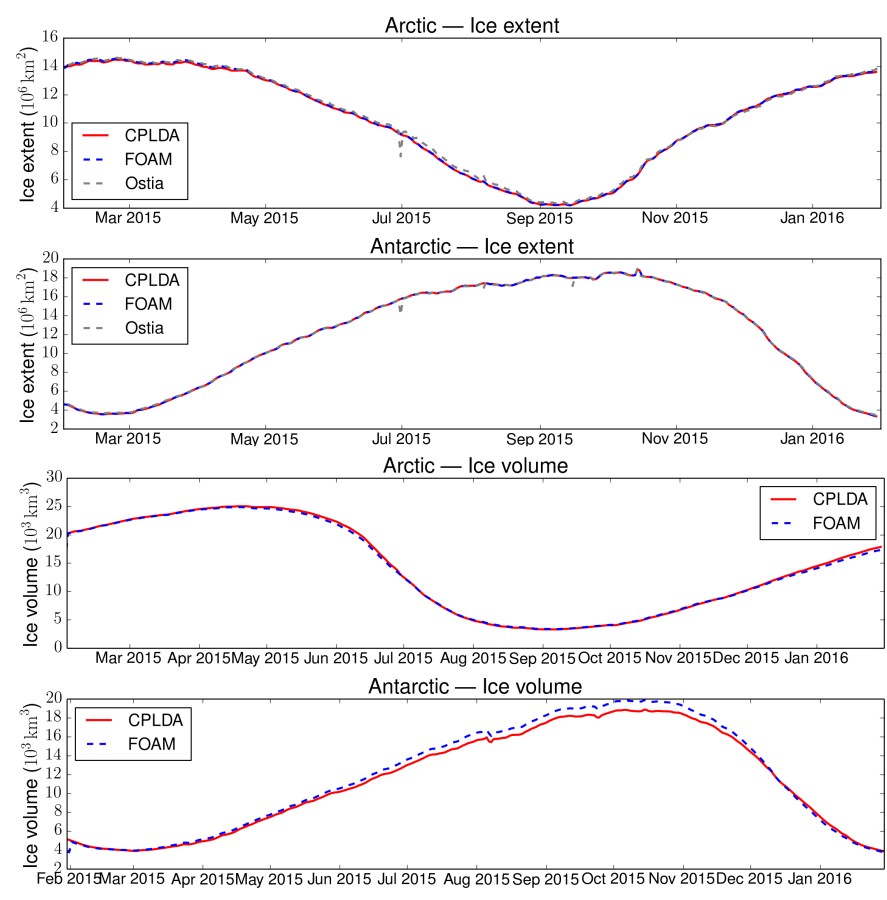

**Figure 3.** Time series of sea ice extent and sea ice volume for CPLDA (2015 experiment) and FOAM

## 3.3 Sea Surface Temperature

We present class 4 statistics against in situ drifting buoy observations provided by US GODAE. The model SST is defined to be the temperature of the top ocean model grid-box which is at 0.5 m depth. The RMSE and mean bias statistics for CPLDA analysis and forecast are shown in Fig 5 and compared to FOAM, PSY4 and GloSea statistics. A small cold bias (-0.02 K) is present in CPLDA in the analysis but does not increase during the forecast. In comparison, the coupled forecast from the GloSea system initialised from uncoupled system exhibits a warm bias significantly increasing during the forecast (+0.09 K after 132 hours; Fig 5a). CPLDA is the only system showing a cold bias, the two uncoupled systems FOAM and PSY4 have a small warm bias. Experiments investigating reducing the assimilation window from 24 to 6 hours in the FOAM system show that the difference observed in bias is unlikely to be due to the shorter assimilation window. The cold bias is therefore more likely related to the differences in heat fluxes between the systems described in section 3.5.

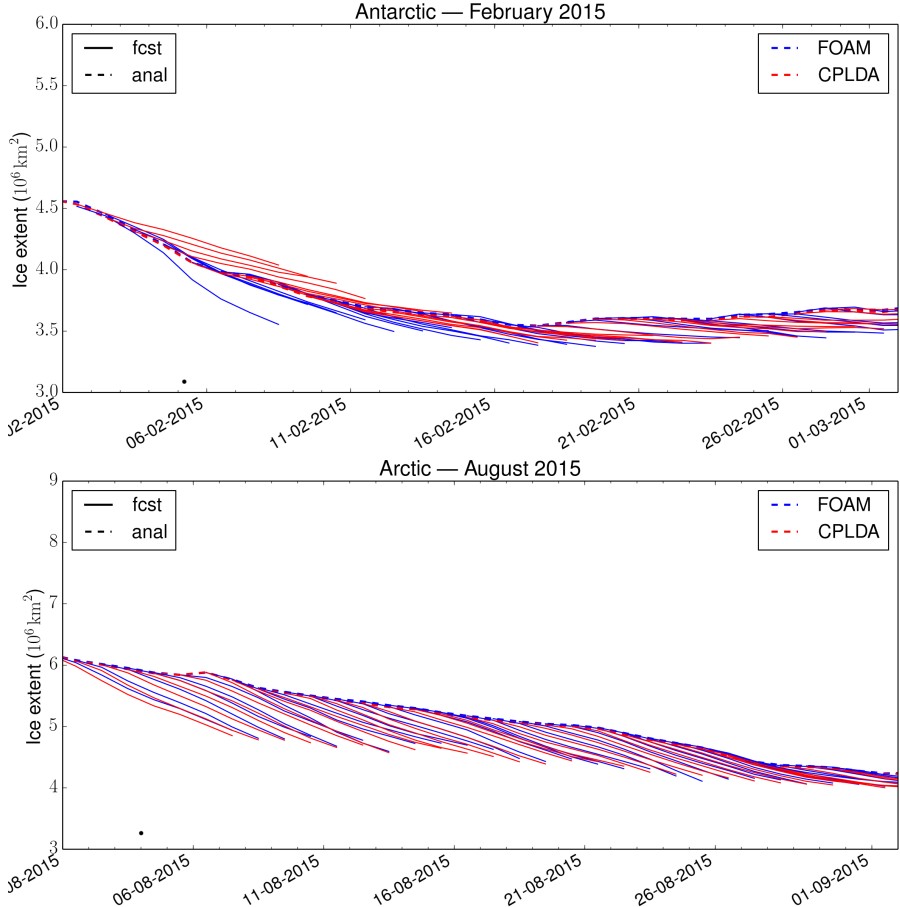

**Figure 4.** Sea ice extent forecast for CPLDA (2015 experiment) and FOAM during the melting season (February in Antarctic and August in Arctic)

The SST RMSE is reduced in the CPLDA analyses compared to the FOAM analyses (this is also observed for the 12-hour forecast). Between 12 and 36 hour forecasts the rate of increase of RMSE is greater in CPLDA than in FOAM so that at forecast length of 36 hours and beyond the RMSE is similar in both systems (Fig 5a). The shorter data assimilation window in CPLDA (6 hours) compared to FOAM (24 hours) explains the improved SST analysis RMSE in CPLDA. Experiments

5 investigating reducing the assimilation window in the FOAM system improved SST analysis statistics (both RMSE and bias). This supports the results of Lea et al. (2015) who suggested that with a shorter cycle the SST model errors have less time to grow. The increase in RMSE between 12- and 36-hour forecasts in CPLDA could be due to the analysis overfitting the observations. The analysis fittingthe observations too closely improves the statistics for the analysis but as a consequence there is a quicker degradation during the forecast. Future work is required to assess the magnitude of this overfitting by withholding

10 a sub-set of observations from the data assimilation (these observations can then be used as an independent validation dataset). The RMSE for analysis SSTs in PSY4 is significantly larger than CPLDA but unlike CPLDA and FOAM, PSY4 does not

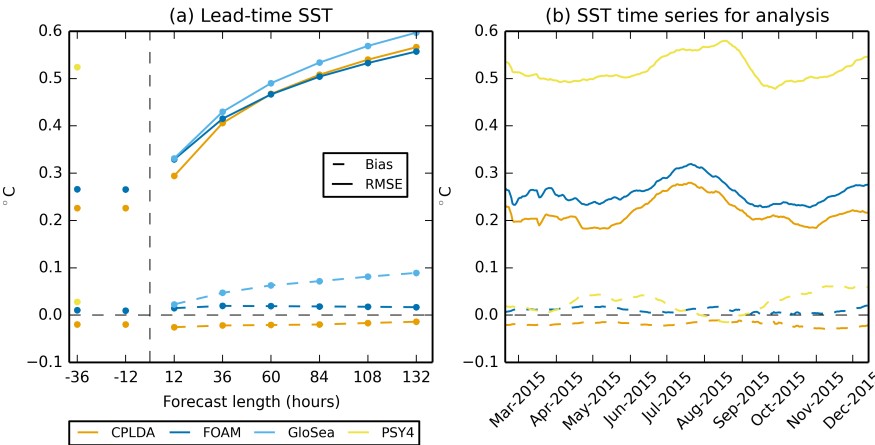

**Figure 5.** SST (model minus observation) class 4 statistics with respect to drifting buoys: (a) RMSE and mean biases at various lead times; (b) Time series of RMSE and mean biases in the "best analysis" from each system.

assimilate SSTs from drifting buoys directly; instead it assimilates the near real-time Operational Sea surface Temperature and sea Ice Analysis (OSTIA) (Donlon et al., 2012), in which the drifting buoys have been assimilated. Furthermore, when using grid point verification PSY4 statistics may suffer from a "double penalty" whereby higher resolution models are often penalised compared to coarser resolution models for missed or misplaced events or false alarms.

The SST RMSE from all models exhibits a seasonal cycle (Fig 5b) with an increased RMSE during the Northern Hemisphere summer. This increase can be attributed to the Northern Hemisphere bias of the observing network in terms of the number of observations. During the Northern Hemisphere summer the mixed layer depth is shallow which leads to a more responsive mixed layer, thus SST variability is increased which is reflected in the increase in the global RMSE during these months.

The CPLDA SST analysis is also assessed via comparison with OSTIA (Fig 6a). The global average CPLDA SST is warmer
than OSTIA, with the difference being 0.06K. This difference corresponds to a known cold bias in the OSTIA analysis relative to independent top-level Argo observations (Roberts-Jones et al., 2012) so could illustrate a potential bias in OSTIA. However CPLDA SSTs (from the top 1 m thick model layer) are also expected to differ from OSTIA because they will capture some of the diurnal cycle while OSTIA is a foundation SST free of diurnal warming. Fig 6b shows the difference between CPLDA and FOAM SST analysis annual means. CPLDA SST is generally colder than FOAM which corresponds to the biases observed in
Fig 5. The differences between the two systems are much smaller than that observed between OSTIA and CPLDA as expected due to similarity in the ocean models and data. Bimodal differences are observed in the areas with high SST gradient such as the Western boundary currents, Antarctic Circumpolar Current and in the Zapiola rise region. These differences are due to differences in the position of SST fronts rather than due to the ability of the two analyses to resolve mesoscale features. To investigate mesoscale feature resolution we calculated the SST spectral power using the method described in Fiedler (2018).
The SST spatial variability for each of the experiments was determined by wavenumber auto spectra analysis. All spectra results were calculated on the native grid of the analysis (ORCA025). For each region of interest, the spectra were calculated

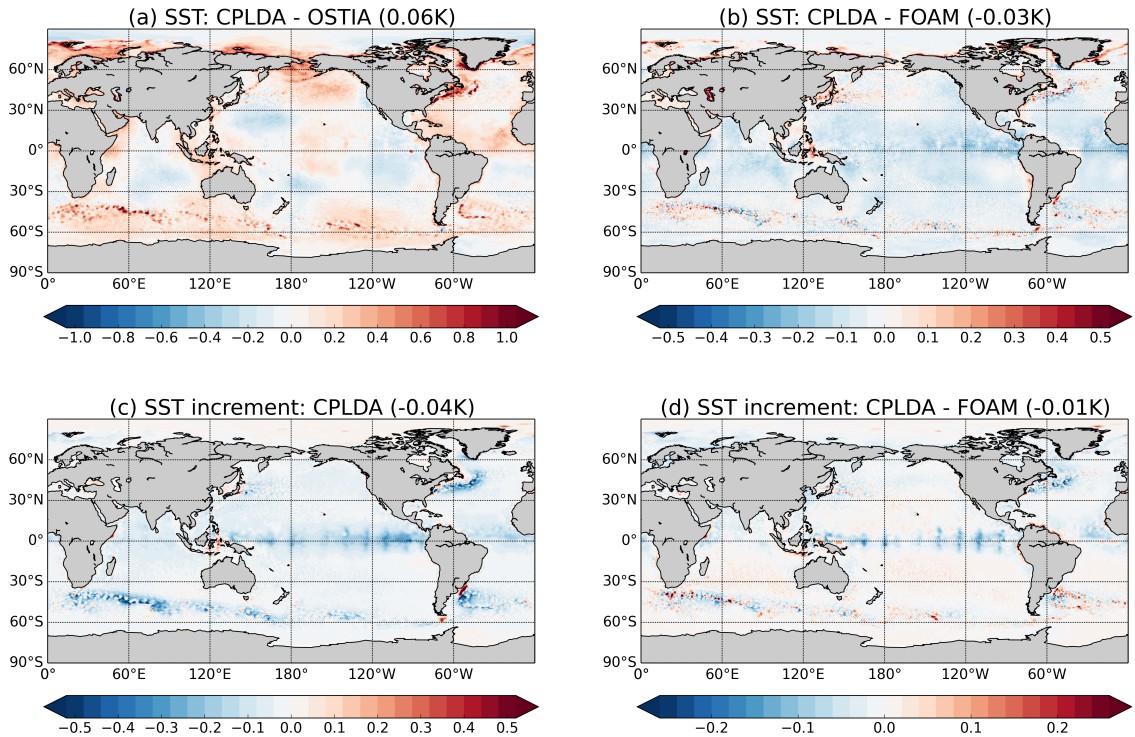

**Figure 6.** CPLDA SSTs in 2015 compared to (a) OSTIA (0.06 K) and (b) FOAM (-0.03 K); (c) mean SST increments in 2015 for CPLDA (-0.04K ); (d) mean SST increments in 2015 for CPLDA compared to FOAM (-0.01 K). Values in parentheses are averages for the global ocean.

along the horizontal coordinates closest to a given latitude. The power spectral density of each latitude was calculated using the Welch's average periodogram method, with 28 degrees of freedom and Hanning window. The overlapping length for each band averaging was half the band length (Fiedler, 2018). The results for CPLDA and FOAM are very similar; in both systems the effective SST resolution is around 50 km.

5    On average, the CPLDA temperature increment at the surface is negative (Fig 6c), meaning that the data assimilation is cooling the model at the surface; this is also an issue in FOAM to a lesser degree and may due to the under-estimation of the wind stress described in section 3.5. The largest increments (both positive and negative) are observed in regions of enhanced SST variability such as the Gulf Stream, Kurushio current and the Antarctic Circumpolar Current. Large negative increments are also applied in the Tropics in regions of tropical instability waves. In the Tropical Pacific, the imprint from the TAO

10  moorings is clearly visible. Fig 6d shows the difference in temperature increment at the surface between CPLDA and FOAM. Bimodal differences are observed in the regions of enhanced SST variability and at the location of the TAO moorings. The degree to which the increments (and therefore the data assimilation) are responsible for the differences in SST between the two systems shown in Fig 5 and Fig 6b varies regionally but large scale similarities are seen in the pattern of the differences.

### 3.4 Temperature and Mixed Layer Depth

The temperature of CPLDA is assessed against Argo profile observations provided by CMEMS (Wehde et al., 2016). The class 4 global temperature statistics for the best analysis are presented in Fig 7a. The results for the forecast (not shown) are similar to the best analysis with the mean bias staying unchanged and the RMSE slightly increasing from 0.67 K at 12 hours to 0.73 K at 108 hours. CPLDA has a cold bias in the subsurface which is maximum (around 0.1 K) at 10 m and present in the thermocline down to approximately 50 m. A smaller warm bias is present at around 100 m, but this is over a greater depth range so represents a large amount of heat. A subsurface cold bias is present in the ocean-only FOAM system but with a smaller amplitude. For the CPLDA results shown, the number of profile observations being assimilated was smaller than in FOAM due to differences in scheduling of assimilation cycles. Even in tests with the operational scheduling (referred to in section 2.6) the sub-surface bias in CPLDA persists despite the increased number of observations being assimilated. In both CPLDA and FOAM the temperature RMSE is largest in the thermocline which highlights its variability; the RMSE is larger in CPLDA than in FOAM.

The temperature profile can be affected by the vertical propagation of surface temperature increments through the water column. King et al. (2018) show that the succession of positive and negative temperature increments has an asymmetric effect on the vertical temperature structure due to the way the temperature increment at the surface is propagated to the bottom of the mixed layer. A negative surface increment weakens the stratification and so deepens the mixed layer, this means that a subsequent positive surface increment is projected deeper. Experiments looking at the assimilation time window in an ocean-only system show that the shorter assimilation cycle in CPLDA (6 hours), relative to FOAM (24 hours), leads to increased temporal noise in the increments and a deepening of the mixed layer.

In CPLDA the large negative increment applied at the surface (Fig 7b) is propagated down to approximately 50 m, below this a small warm increment is applied down to approximately 150 m. The fact that the negative increment is projected deeper in CPLDA than in FOAM and the dipolar structure in the vertical is consistent with idealised experiments into vertical propagation of temperature increments, (King, pers. comm. 2018). CPLDA has a larger negative increment at the surface than FOAM. Below approximately 200 m the magnitude of the average temperature increment is small (Fig 7b) and the increments applied by CPLDA and FOAM are similar. The differences observed in global average temperature increments are reproduced in the ocean-only assimilation time-window experiments comparing 6- to a 24-hour window. It is therefore likely that the differences observed in the global temperature increment are due to the shorter assimilation window in CPLDA.

Mixed Layer Depth statistics from the analysis confirm that CPLDA has a deeper MLD than the assimilated profile observations. The MLD statistics presented here use the Kara mixed layer depth with a density based criteria. CPLDA mean error against the assimilated observations is 5.2 m too deep while the RMSE is 34.7 m. Similarly the MLD in FOAM is deeper than the observations but both the mean error (2.1 m too deep) and the RMSE (32.6 m) are reduced. These statistics confirm that the CPLDA MLD is deeper than FOAM (Fig 8). During the forecast MLD RMSE and bias persist for both systems (not shown). Experiments running FOAM with a 6-hour assimilation time window show that the shorter window results in an over-deepening of the MLD by 2.9 m relative to that using a 24 hour window; this is consistent with the 3.1 m difference observed

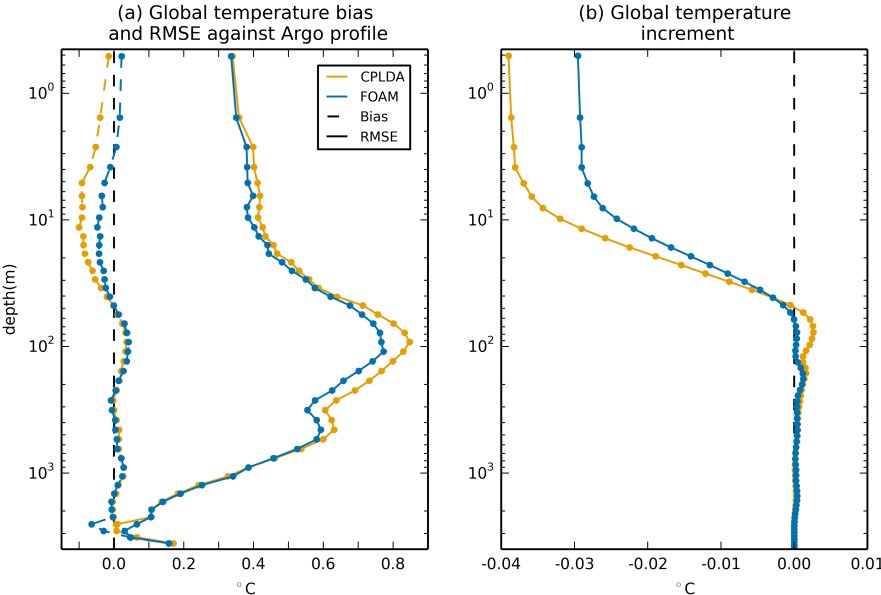

**Figure 7.** (a) CPLDA and FOAM global average mean bias and RMSE temperatures compared to Argo observations using class 4 methodology; (b) Global average mean temperature increments applied in CPLDA and FOAM.

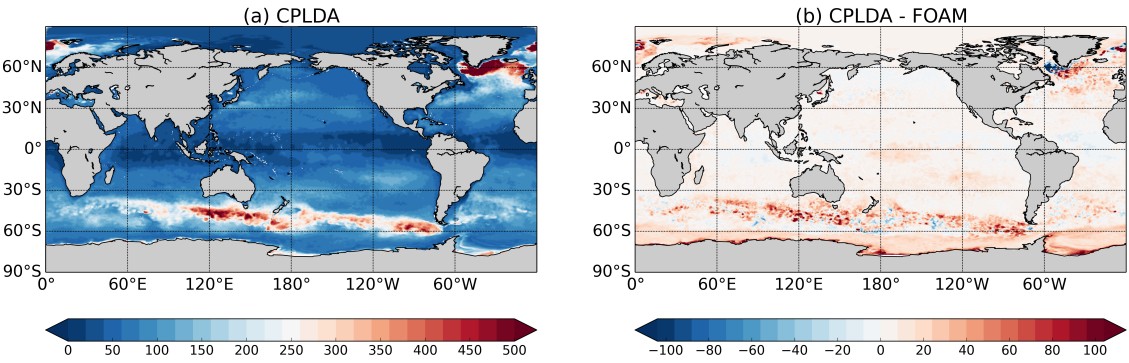

**Figure 8.** Annual mean Kara mixed layer depth in m (Kara et al. (2000) using temperature criterion of $0.8°C$: (a) CPLDA, and (b) difference between CPLDA and FOAM. Positive value means CPLDA MLD is deeper than FOAM MLD.

between CPLDA and FOAM. The change in MLD due to the shorter assimilation window may be caused by the asymmetric effect of the surface temperature increment on vertical temperature structure. The over-deepening of the mixed layer will lead to a misplacement of the thermocline which will contribute to the larger RMSE in the thermocline in CPLDA than in FOAM observed in figure 7a. It is worth noting that differences in wind stress between CPLDA and FOAM will also affect the MLD.

5    Experiments support the shorter assimilation window in CPLDA causing the differences observed in the global increments (Fig 7b) and in MLD (Fig 8) between CPLDA and FOAM. However these experiments were not able to reproduce the sub-

surface bias observed in CPLDA relative to FOAM (Fig 7a). We are currently unable to attribute this bias to the shorter assimilation window. This may be due to the experiments not being long enough (6 months) or more likely that the bias is a result of the differences in forcing between CPLDA and FOAM described in the next section. Future work is planned to look in depth at the vertical propagation of surface increments using a 1-D model and may aid in the attribution of the observed

sub-surface bias.

## 3.5 Atmosphere and surface fluxes

In this section, we assess the atmosphere component of CPLDA, focussing on the interface with the ocean, in particular the surface fluxes. We compare CPLDA surface fluxes to FOAM over the ocean; we have not yet investigated the fluxes over the ice. The differences in net total heat flux are shown in Fig 9. On average CPLDA receives less heat than FOAM (-3.9 W m$^{-2}$).

CPLDA receives less (or loses more) heat in the Tropics than FOAM except in the Gulf of Guinea and in the East Pacific along the Peruvian and Chilean coast. At higher latitude, CPLDA loses less heat than FOAM especially in the regions of large latent heat loss (Kuroshio and Gulf Stream). The reduced heat gain compared to FOAM between 30S and 30N contributes to the colder SST observed in CPLDA. The differences in net heat flux compared to FOAM are mainly due to differences in short-wave radiation and particularly in latent heat (Fig 9). The small scale pattern differences are due to differences in the

representation of the mesoscale between the two models (eddies are not completely constrained by the observatiosn and so are not located in exactly the same positions).

Fig 9c shows the difference in short-wave radiation between CPLDA and FOAM. On average, CPLDA receives more short-wave radiation than FOAM (+1.05 W m$^{-2}$). The difference is large in the southern hemisphere where CPLDA receives significantly less short-wave radiation than FOAM. In the equatorial region, CPLDA receives less short-wave radiation in the West

Pacific but receives significantly more short-wave radiation than FOAM in the East Pacific and in the Atlantic including the whole Gulf of Guinea. The stratocumulus clouds are under-estimated in the Met Office atmospheric model and cause a short-wave over-estimation on the eastern boundary upwelling systems. The differences in short-wave radiation between CPLDA and FOAM in these regions suggest that this bias is larger in CPLDA. In the northern hemisphere, CPLDA receives less short-wave radiation in the Pacific, while in the Atlantic it receives more. Overall, the differences observed in short-wave radiation

are significant and contribute to the differences observed in total heat fluxes. We have performed a shorter simulation of the CPLDA system with a higher atmospheric resolution (17 km) which gives similar results (not shown) to CPLDA meaning that the differences in the fluxes between CPLDA and FOAM are not caused by the differences in resolution.

The differences in latent heat can be decomposed into two components. First, the atmosphere component in CPLDA differs from the atmosphere model (NWP) used to force FOAM. Fig 9d shows the difference between CPLDA and NWP latent heat.

The NWP latent heat is calculated using the same bulk formulae as CPLDA but is using OSTIA SSTs as a surface boundary condition. This shows that even with the same bulk formulae the evaporation is significantly higher in CPLDA than in NWP. In CPLDA, the air at 10 m is drier than in NWP causing increased evaporation and increased latent heat loss. The signal is stronger in the Tropics and contributes to the differences in total heat flux seen between CPLDA and FOAM, with FOAM losing less heat than CPLDA in the Tropics.

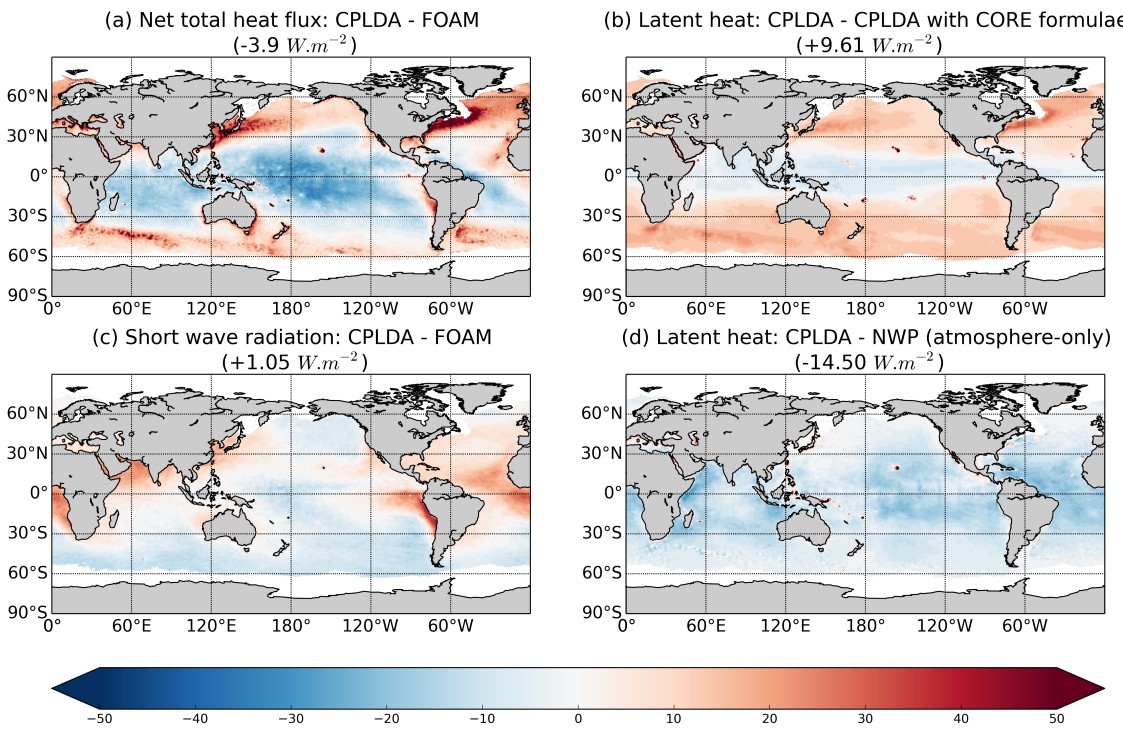

**Figure 9.** Annual mean net downward heat flux differences, in W m$^{-2}$ : (a) total heat flux difference between CPLDA and FOAM (-3.9 W m$^{-2}$), (b) latent heat flux difference caused by using COARE3.0 bulk formulae in CPLDA rather than CORE as in FOAM (+9.61 W m$^{-2}$), (c) short-wave radiation difference between CPLDA and FOAM (+1.05 W m$^{-2}$), and (d) latent heat difference between CPLDA and the atmosphere-only NWP system, both of which are using COARE3.0 bulk formulae (-14.5 W m$^{-2}$). In bracket, are the average values for the global ocean.

Secondly, there are differences in latent heat due to the use of different bulk formulae in CPLDA and FOAM. To investigate the difference caused by the bulk formulae, we recalculated CPLDA fluxes using CORE bulk formulae (Large and Yeager, 2004) as used in FOAM. Fig 9b shows the differences in latent heat when using CORE formulae. At high latitude the latent heat loss is increased with the largest differences observed in the regions with large latent heat loss. On the other hand, in 5    the equatorial band, the latent heat loss is reduced with CORE formulae. The impact of the bulk formulae on the latent heat calculation is significant and contributes to the differences in total heat flux. An impact is also observed on the sensible heat but with a much small magnitude (not shown). The addition of the differences in latent heat caused by having a different atmospheric state (-14.5 W m$^{-2}$ Fig 9d) and those caused by using different bulk formulae (+9.61 W m$^{-2}$ Fig 9b) can explain much of the difference seen in the net heat flux into the ocean between CPLDA and FOAM (Fig 9a).

10    The sum of the latent heat differences and short wave differences equals -3.84 W m$^{-2}$ , close to the difference in net total heat flux (-3.9 W m$^{-2}$).

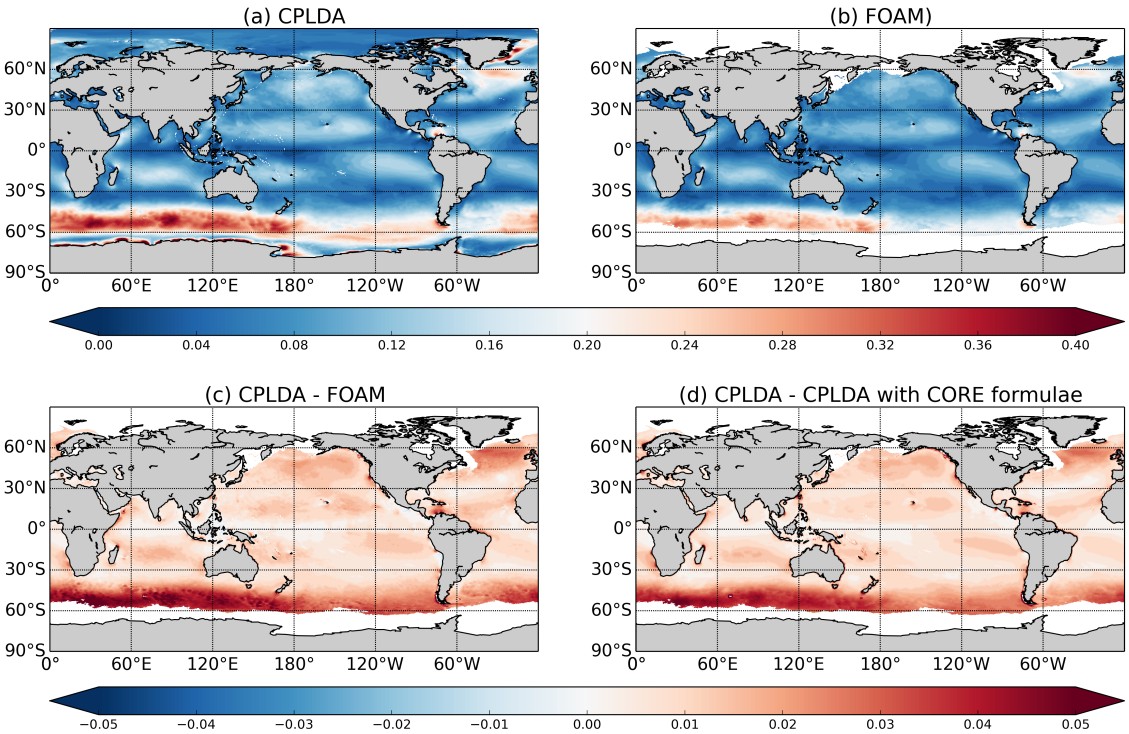

**Figure 10.** Annual mean wind stress magnitude in N m$^{-2}$: (a) CPLDA, (b) FOAM, (c) Wind stress difference between CPLDA and FOAM, and (d) Wind stress difference caused by using COARE3.0 bulk formulae in CPLDA rather than CORE as in FOAM

Air-sea momentum flux is a crucial forcing for the ocean as it is through the wind stress that the atmosphere drives the ocean. The CPLDA wind stress magnitude annual mean is shown in Fig 10a). The wind stress in CPLDA is almost everywhere stronger than in FOAM. The differences in wind stress between CPLDA and FOAM are mainly caused by the different bulk formulae. Indeed, when recalculating CPLDA wind stress using the same (CORE) bulk formulae as FOAM, the differences
5    between CPLDA and the new calculated wind stress are similar to the differences between CPLDA and FOAM (Fig 10c and 10d). The new wind stresses are calculated with CORE bulk formulae but using CPLDA windspeed and surface currents. This highlights the importance in the choice of bulk formulae for the wind stress calculation. However, we note that it is not equivalent to running CPLDA using CORE bulk formulae, as reduced wind stress would imply feedback to the atmosphere causing an increase in wind speed and therefore limiting the reduction in wind stress.
10    Because of the large differences caused by the bulk formulae, comparing the differences in wind stress is equivalent to comparing the different drag coefficient used. Brodeau et al. (2017) highlighted disagreement in drag coefficient between different bulk formulae with the COARE 3.0 algorithm producing higher wind stress. They mention that the latest improvements in the COARE algorithm (version 3.5) suggest that the drag coefficient of COARE3.0 is likely too small in strong wind conditions.

This suggests that the wind stresses calculated in CPLDA may be underestimated at times, although improved compared to those used by FOAM.

## 3.6  Velocities

Predictions of ocean currents are important for marine activities. They are used for a number of practical applications such as ship routing, marine search and rescue, pollution monitoring, the offshore oil and gas industry, and marine renewable energy. To assess the velocities at 15 m, we compare CPLDA to measurements from drifters. The observations are independent as no ocean current observations are assimilated by the system. We use an in situ delayed mode product from CMEMS (Etienne, 2017); this is designed for reanalysis purposes with the best available version of in situ data for ocean surface currents. The data are collected from the Surface Drifter Data Assembly Centre (SD-DAC at NOAA AOML). All surface drifters data are processed to check for drogue loss and a wind slippage correction is applied to undrogued buoys. The wind slip correction is computed following Rio (2012). We compared 15 m model velocities against 15 m CMEMS observations corrected with wind slippage for the year 2015. Despite a limited number of observations compared to SST observations, the observation coverage from drifters is generally good except in the Tropical Atlantic.

The class 4 statistics for the 15 m velocities for the global ocean are presented in Fig 11. On average the CPLDA analysis has a negative bias (-0.07 m s$^{-1}$) and a RMSE of 0.19 m s$^{-1}$. In all regions except the Tropical Pacific, both the bias and the RMSE are stable during the forecast (not shown). The bias in the uncoupled system analyses (FOAM and PSY4) is the same as CPLDA and the RMSE close (0.18 for FOAM; 0.17 for PSY4). The larger RMSE in CPLDA analysis is mainly due to periods of large RMSE in April 2015 and November 2015 in the Tropical Pacific (see Fig 11). In regions other than the Tropical Pacific no significant differences are observed, with CPDLA and FOAM statistics similar. The increased RMSE in the analysis is caused by spurious currents in the West Tropical Pacific (Fig 12). During the forecast, the currents weaken and the RMSE decreases suggesting they are caused by the data assimilation. The shorter data assimilation window in CPLDA limits the number of observations and causes noisier increments; in addition a reduced total number of SLA observations are assimilated in CPLDA compared to FOAM. We re-ran the CPLDA system for a month from 15/10/2015 with the updated scheduling allowing more observations to be assimilated. In this experiment, the RMSE in the analysis is significantly reduced and now similar to FOAM and PSY4 (see shaded area in Fig 11). The spurious currents north of Indonesia are suppressed (Fig 12). However, in the current CPLDA operational system (running with the updated scheduling allowing more observations to be assimilated), there are still periods with unrealistic currents developing in the West Tropical Pacific (not shown). These unrealistic currents caused by SLA assimilation are not present in the FOAM system which uses the same assimilation scheme and assimilates the same observations as CPLDA. This strongly suggests that they are caused by the shorter assimilation window in CPLDA.

Tuning the error covariances for the shorter time window may reduce or eliminate the above problem. Smaller estimated background errors would give the observation less weight in the analysis and lead to smaller, less noisy increments. Future work investigating re-estimating the background error covariances is planned. The updated CNES-CLS13 Mean Dynamic Topography of Rio et al. (2014) is significantly improved, particularly around the Maritime Continent, compared to the CNES-CLS09 version used in these experiments. Hence using a newer MDT in the CPLDA system may also reduce this issue.

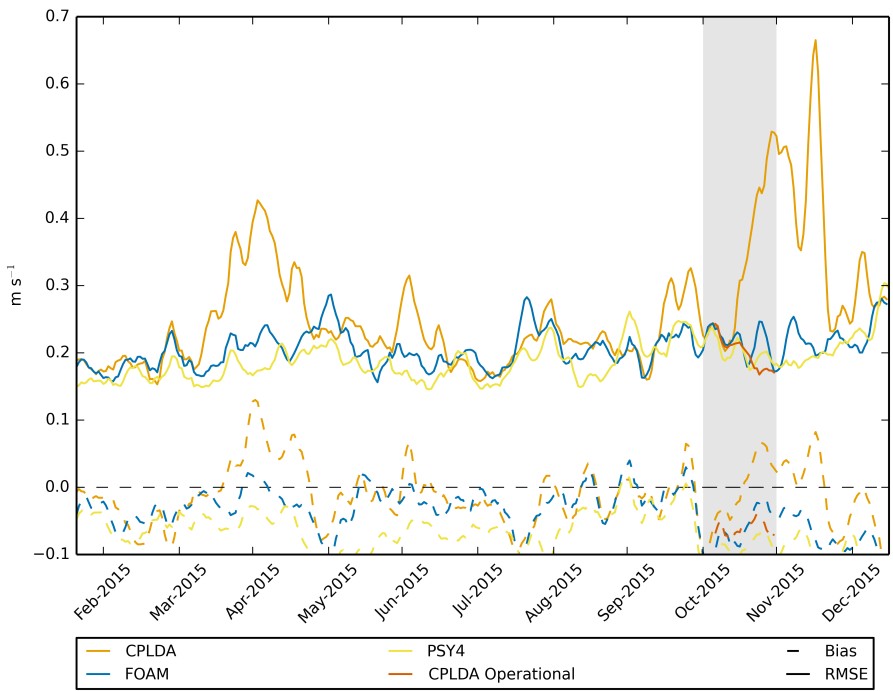

**Figure 11.** 15 m velocity using class 4 statistics with respect to velocities derived from drifting buoys for 2015. RMSE and mean bias are shown for CPLDA and FOAM, as well as PSY4 and, for a short test period, CPLDA using the operational scheduling allowing assimilation of additional SLA observations.

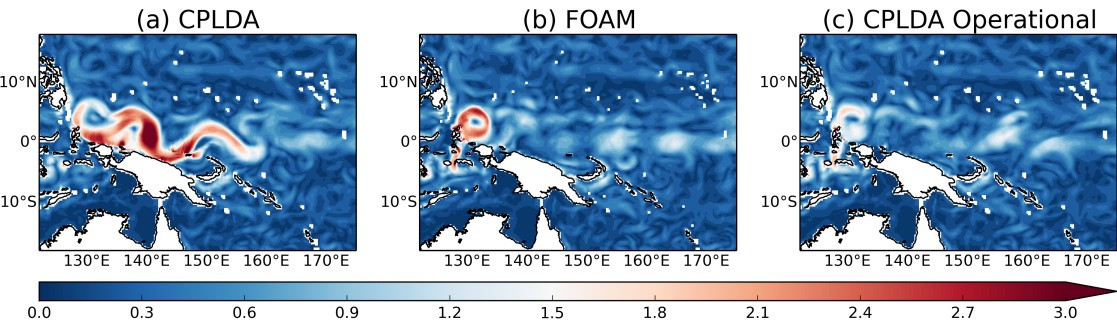

**Figure 12.** 15 m velocity average (in m s$^{-1}$) in the West Tropical Pacific for the week from 05/11/2015 to 11/11/2015 for (a) CPLDA, (b) FOAM and (c) CPLDA with the current operational scheduling allowing assimilation of additional SLA observations.

Despite the mean bias and RMSE values, CPLDA velocities are only moderately correlated to observations (0.50). This correlation is weaker than in FOAM (0.55) and PSY4 (0.62). Only PSY4 is skillful using the definition that a correlation greater than 0.6 is an indication of a skillful forecast (Murphy and Epstein, 1989; Hollingsworth et al., 1980). Correlation varies substantially from region to region (Table 2). In some regions, CPLDA correlation is near or above 0.6, as in the North

Pacific, while in North Atlantic the correlation is poor (0.10). All the models have skillful correlation in the North Pacific but poor correlation in the North Atlantic. This was previously observed by Blockley et al. (2012) in an earlier version of the FOAM system. They explained the lack of skill in the North Atlantic by the domination of the mesoscale which is more difficult to predict while in more benign regions the model performed better. It is in the Southern Ocean, a region largely dominated by the mesoscale, that the $1/12°$ model (PSY4) most clearly outperforms the $1/4°$ models (CPLDA, FOAM). Here PSY4 has a correlation of 0.62 while CPLDA and FOAM have correlations of 0.45 and 0.50 respectively.

Eddy Kinetic Energy (EKE) was also compared in CPLDA, FOAM and PSY4. In an ocean-only model, using the ocean velocity to calculate the wind stress has a damping effect on the eddies (Duhaut and Straub, 2006; Dawe and Thompson, 2006; Renault et al., 2016). Despite the expected reduced damping effect in a coupled system, we did not observe a higher EKE in CPLDA than in FOAM. However at $1/4°$ resolution, the mesoscale is poorly represented and most of the EKE is injected into the model by the SLA assimilation. Hence differences in EKE between CPLDA and FOAM are mainly caused by differences in SLA observations assimilated, or impacts of the shorter assimilation window, rather than due to a reduced eddy damping by the wind stress.

## 4    Conclusions

Since July 2017, the CPLDA system has been delivering an ocean analysis and forecast to CMEMS operationally. This was a significant upgrade as prior to this ocean products were delivered to CMEMS from the coupled GloSea system which was initialised from uncoupled ocean (FOAM) and atmosphere (Met Office NWP) analyses. Here we have assessed the first long (1-year) experiment using the CPLDA system in a pseudo-operational mode. Previous studies with an earlier similar system (Lea et al., 2015) presented the results only from short (1-month) trials. This longer trial has allowed an in-depth assessment of the CPLDA system against observations, and comparison of its ocean analaysis and forecast to current benchmark products like the Met Office $1/4°$ model FOAM and the Mercator $1/12°$ model PSY4.

Overall, the CPLDA system performs well compared to the FOAM system. After applying an update to the scheduling to allow more observations to be available at the run-time, the SLA statistics are now similar to FOAM statistics both for the bias and RMSE. The SST statistics are improved with an RMSE significantly smaller than in FOAM and the warm bias that was developing in the coupled GloSea forecasts is not present in CPLDA. However, despite improving the SST, the vertical propagation of the SST increments may have contributed to an increased sub-surface cold bias in CPLDA. The cold bias is present in FOAM but is worsened in CPLDA. The increased cold bias is also associated with a deepening of the MLD in CPLDA. The statistics for the 15 m current are similar in CPLDA and FOAM except for two periods when CPLDA exhibits large errors in the West Tropical Pacific caused by the SLA assimilation. Again these are largely addressed by the scheduling modification to allow more observations to be assimilated. The 15 m currents from both models have a poor correlation to observations.

Compared to other systems used to produce ocean analyses, the short data assimilation window distinguishes the CPLDA system. The impact of the data assimilation time window was investigated by Lea et al. (2015) (1 month experiment) and

further investigated in experiments described herein (6 month experiments). The FOAM system was run with a 6-hourly cycle and assimilation window and compared the results with the standard FOAM 24-hourly cycle.Lea et al. (2015) found a small impact on the temperature and salinity profiles, and on the SLA statistics, but saw an improvement in the SST statistics explained because with a shorter cycle the model errors have less time to grow. These results were supported by the new

longer experiments detailed in this paper. The shorter window in CPLDA compared to FOAM has a positive impact on the SST analysis but the RMSE increase during the forecast is enhanced in CPLDA suggesting that the system may be overfitting the observations. The shorter assimilation window in CPLDA also leads to the surface temperature increments being propogated deeper and to an over-deepening of the mixed layer depth. This can be partly explained by the asymmetric effect of noisy surface temperature increments on the vertical temperature structure. In the assimilation scheme the depth to which the SST

increment is propagated is determined by the Mixed Layer Depth of the background field. In the current system, the background field used is that at the first time step of the observation operator. To reduce the noise, a future improvement could be to use a daily mean field instead of the instantaneous field. This method has been tested with a UK shelf model (King et al., 2018) but has not yet been tested for the CPLDA system.

The impact of the short assimilation is also seen in the surface currents near the equator where the SLA assimilation is

responsible for the large erroneous currents in CPLDA in the West Tropical Pacific. The problem is also present in the FOAM system with a 24-hour window; indeed both systems have excessive Eddy Kinetic Energy along the equator. However, it is likely that having a 6-hourly cycle makes the problem worse as it limits the numbers of observations assimilated creating large increments which cause the spurious currents. It highlights the need to improve the SLA assimilation, in particular, using background error covariances appropriate to a 6-hour assimilation window. The error covariances currently used are the same

as those used the FOAM system, but estimates specific to the CPLDA system would take into account the 6-hour assimilation window as well as the different model error characteristics of the coupled model relative to the ocean-only model. The 1-year run carried out gives us the data to allow the estimation of error covariances for the CPLDA system; making these improvements to the ocean DA should help to resolve the problems seen in equatorial currents and hopefully improve the correlation of the model velocity to drifter velocities.

For many years, ocean analyses and forecasts have been produced from forced ocean models. Using a coupled system brings some new challenges. When assessing the CPLDA system it is important to understand which changes are genuine impacts of coupled processes and feedbacks, as opposed to unavoidable changes in the system set-up (like, for example, the use of a 6-hourly assimilation window). The comparison between CPLDA and FOAM surface fluxes highlighted some significant differences in CPLDA and NWP atmospheric fields. Firstly, there is a significant difference in 10 m air temperature with the

NWP fields (used to force FOAM) being warmer than CPLDA; this is also associated with an increased specific humidity at 10 m. Differences in both air temperature and humidity are large scale and especially significant between $30°$S and $30°$N. With colder and drier air at 10 m in CPLDA, the evaporation is larger causing more heat loss by latent heat. This extra heat loss in CPLDA compared to FOAM contributes to the differences in SST observed with CPLDA being colder than FOAM. Given we have been unable to attribute the sub-surface bias observed in CPLDA to the shorter assimilation window, it is likely this is

also related to the surface flux differences between CPLDA and FOAM.

The differences in atmospheric configurations between the NWP model used to force FOAM in 2015 and that in CPLDA mean it is difficult to separate the impact of the coupling. A more in-depth investigation with a recently developed NWP model using the same configuration as CPLDA is required to understand how much of these differences are caused by the coupling. In addition, further work is needed to understand the relationship between the differences in short-wave radiation and those in surface air temperature and specific humidity.

The assessment of the surface fluxes also emphasised the importance of the bulk formulae. In CPLDA, the fluxes are calculated by the atmosphere component using the UM bulk formulae (based on COARE3.0) while in FOAM the fluxes are calculated by NEMO using CORE bulk formulae. One of the main impacts of the different bulk formulae is the differences in wind stress. Despite having similar winds, the magnitude of the CPLDA wind stress is significantly larger than that of FOAM, particularly in the Southern Ocean. This is partly responsible for the increased, and too deep, Mixed Layer Depth in CPLDA.

The 40 km atmospheric resolution of CPLDA is not high enough to make sensible comparisons of NWP performance with that of the Met Office operational NWP system (which at the time had a resolution of 17 km) but a basket of metrics used for assessing model performance suggests that CPLDA atmospheric performance is good and at least comparable to atmosphere-only systems at equivalent resolution. In fact because the 2015 CPLDA trial included VarBC (Cameron and Bell, 2016) which was not included in the NWP system until 2016, some aspects of the CPLDA system out-performed the much higher resolution NWP system as operational at the time.

A "coupled NWP" system is now being developed based upon the operational CMEMS system described here. This will have a much higher resolution (10 km) atmosphere with the aim of delivering both weather and ocean forecast products from a single system by 2021. On this timescale it is hoped that it will be possible to address some of the issues already discussed where the CPLDA performance is slightly degraded compared to FOAM. However the very good performance for analysis SSTs in CPLDA compared to both FOAM and OSTIA, as well as the ability to evolve these through the forecast, suggests that such a system will be well-placed to improve upon the performance of the existing NWP system. A subsequent upgrade of this system would be to increase the ocean resolution to $1/12°$ as well as incorporating a wave model and using ensemble information to improve the ocean data assimilation.

*Code availability.* The Met Office Unified Model (MetUM) is available for use under licence. A number of research organizations and national meteorological services use the UM in collaboration with the Met Office to undertake basic atmospheric process research, produce forecasts, develop the UM code, and build and evaluate Earth system models. For further information on how to apply for a licence, see http://www.metoffice.gov.uk/research/modelling-systems/unified-model.

5      JULES is available under licence free of charge. Further information on how to gain permission to use JULES for research purposes can be found at https://jules-lsm.github.io/.

The model code for NEMO v3.6 is available from the NEMO website (http://www.nemo-ocean.eu). On registering, individuals can access the code using the open-source subversion software (http://subversion.apache.org/).

The model code for CICE is available from the Met Office code repository https://code.metoffice.gov.uk/trac/cice/browser. In order to

10     implement the scientific configuration of GC2 and to allow the components to work together, a number of branches (code changes) are applied to the above codes. Please contact the authors for more information on these branches and how to obtain them.

*Competing interests.* No competing interests are present.

*Acknowledgements.* This work has been carried out as part of the Copernicus Marine Environment Monitoring Service (CMEMS). CMEMS is implemented by Mercator Ocean International in the framework of a delegation agreement with the European Union.

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

**Table 1.** The FOAM , CPLDA and GloSea ocean configurations

| Ocean configuration | FOAM | CPLDA | GloSea forecast |
|---|---|---|---|
| Data assimilation | NEMOVAR (3D-Var-FGAT, dual length scale) with 40 iterations | NEMOVAR (3D-Var-FGAT, dual length scale) with 40 iterations | N/A |
| Assimilation time window | 24h | 6h | N/A |
| Surface forcing | CORE and CICE bulk formulae with UKMO GA6.1 NWP fields | Directly coupled (COARE3.0) GA6.0 fluxes | Directly coupled GA6.0 fluxes |
| Surface forcing resolution | 17km (3-hourly heat fluxes and 1-hourly 10m wind) | 40km resolution (1-hourly) | 50km resolution (3-hourly) |
| Penetration radiation | R-G-B (Lengaigne et al., 2007) | 2-band (Paulson and Simpson, 1977) | 2-band (Paulson and Simpson, 1977) |
| Rivers | Climatological estimates (Bourdallé-Badie and Treguier, 2006) | Climatological estimates (Bourdallé-Badie and Treguier, 2006) | Calculated by river scheme in Unified Model |
| Haney retroaction | SSS (-33.33 mm day$^{-1}$ psu$^{-1}$) | SSS (-33.33 mm day$^{-1}$ psu$^{-1}$) | None |
| 3D Newtonian damping | Temperature and salinity (1-year time scale) | Temperature and salinity (1-year timescale) | None |
| Pressure gradient correction | Yes (Bell et al., 2004) | Yes (Bell et al., 2004) | None |
| Salinity dependent freezing temperature | Yes | No | No |

**Table 2.** Velocity class 4 statistics against drifting buoys for 2015 in m s$^{-1}$

| Model | CPLDA | | | FOAM | | | PSY4 | | |
|---|---|---|---|---|---|---|---|---|---|
| | Bias | RMSE | Correlation | Bias | RMSE | Correlation | Bias | RMSE | Correlation |
| Global | -0.07 | 0.19 | 0.50 | -0.07 | 0.18 | 0.55 | -0.07 | 0.17 | 0.62 |
| North Atlantic | -0.15 | 0.21 | 0.10 | -0.14 | 0.21 | 0.12 | -0.15 | 0.21 | 0.14 |
| North Pacific | -0.06 | 0.58 | 0.60 | -0.06 | 0.58 | 0.62 | -0.06 | 0.58 | 0.63 |
| Southern Ocean | -0.09 | 0.25 | 0.45 | -0.08 | 0.24 | 0.50 | -0.08 | 0.22 | 0.62 |