# Peer review of "Assessment of ocean analysis and forecast from an atmosphere-ocean coupled data assimilation operational system"

_Ocean Science, 2018_

## Referee Comment (RC1) · Anonymous Referee #1 · 4 Mar 2019

General comments:

This study presents a new ocean-atmosphere coupled operational system (CPLDA) with weakly coupled data assimilation developed at Met Office. In the first part, the model components are described and compared to previous coupled and uncoupled systems (GloSea and FOAM respectively). The weakly coupled data assimilation method from Lea et al. (2013, 2015) is also detailed. This system is operated and evaluated during one year (2015) with 6-hour analysis and daily 7-day forecast. The system performances are then compared with the operational ocean-only FOAM (Ryan et al. 2015) and Mercator PSY4 systems and with different observational (SST, MLD,

15 m velocities, SLA) datasets. Atmospheric forcings and turbulent fluxes are also analysed. In summary, the CPLDA system performs as well as ocean-only systems despite its increased complexity and constitute a promising first step toward a fully coupled operational system.

The manuscript provides a precise description of the CPLDA system, underlining the benefits and limitations compared to other MetOffice systems and observations. It is globally well written and scientific analyses are seriously presented. However, I recommend the following modifications to improve the manuscript quality and understanding. I have the feeling that some sensitivity tests and analysis are still missing to better cover the system description and validation. This is mainly due to the fact that besides the ocean model similar configurations between CPLDA and FOAM, there a large number of differences between the two systems. Consequently, it is sometimes difficult for the authors to assess precisely the differences between the systems because too much parameters change at the same time. Some additional sensitivity tests allow to better understand those differences (for example, the one-month experiment is a different scheduling for the DA). But others sensitivity tests are missing to clearly understand and to disentangle the numerous modifications between the systems. As a consequence, some analysis are not very convincing because of this lack of sensitivity experiments. Depending on the authors will and capacity to run these additional simulations, some corrections will have to be made to the text to confirm or infirm the hypothesis proposed. If no supplementary experiments can be done, some comments must be added to the text to emphasize the limitations of the study. Because atmospheric forcings and surface fluxes are strongly related to SST, upper temperature and MLD, the section 3.4 should be moved before the section 3.3 about velocities (that I would rename "upper or ML velocities"). It is not always clear if the authors are talking about the analysis, the forecast or both in the whole manuscript. It makes the study understanding less clear to follow and more difficult to understand. I think the manuscript can be easily improved by clearly stating this.

Specific comments:

p.1 - Introduction: The introduction is not really relevant because it is mostly a repeat of the system description in section 2. The introduction should be improved to better describe the context of this study, i.e to describe the main physical and technical arguments in favour of developing a coupled system with weakly coupled assimilation (instead of independent systems and assimilations). A description of what is done in other operational centres (ECMWF, NCEP, . . .) could also help to better understand the framework and the interest of the system presented here.

p.3 l.10-12: it is not clear if VarBC is activated or not in the present study, please specify this in the text.

p.5 l.30: please justify why such changes regarding IAU and observation operator were made in CPLDA compared to FOAM.

p.6 l.20: scheduling of the operational system ?

p.6 l.25: I don't understand why some results should come from the operational version of the system and others from "test" version ? I find this confusing. Perhaps it should be better to talk about the actual operational version only in the conclusion to avoid any confusion.

p.6 l.31: please add Lellouche et. al 2018 as a reference for Mercator-Ocean PSY4 system.

p7 l.9-11: I think these results are important and should be added to the paper as a distinct section or paragraph with a dedicated figure to illustrate the differences in terms of SLA bias and RMSE between FOAM and the 2 versions of CPLDA with the different schedulings. It is also unclear if the results presented here are those from the current operational system or from the version used in the study ("old scheduling").

p.14 section 3.4 should be moved close to the sections 3.1 and 3.2 as it is directly related to changes in SST and MLD. It would improve the readability of the manuscript.

p.7 l.28-30 This hypothesis should be directly tested by doing some additional sensitivity experiments using a 24h window with CPLDA or a 6h window with FOAM to confirm it or not.

p.7 l.30 "in CPLDA" -> "in CPLDA analysis": it is not clear when the author is talking about the analysis or the forecast. Please specify it everywhere it is necessary in the text.

p.7 l.31: is there a way to measure this "overfitting" ? It is not clear why FOAM SST forecast performs better than CPLDA and how is it related to this overfitting.

p.8 fig.1: the forecast duration is 7 days (168h), please expand the figure axis accordingly

p.8 l.19: it could be interesting to give the effective SST resolution for both system even without showing the figure to get an idea of the scale range resolved.

p.9 section 3.2: please add PSY4 in the analysis and Figure 3 if possible to be more coherent with others sections and to get a more complete comparison between products.

p.9 l.9: please give the reference associated with this product.

p.9 l.11: please give the range of the RMSE increase to be able to compare it to the SST RMSE.

p.10 l.15: the warm bias and associated significant RMSE ($\sim$1°C) located at $\sim$100m is not discussed nor explained in the text. Please add a paragraph about this. If the discussion about King et al. 2018 results explain it, it should appear more explicitly in the text.

p.10 l.17: are you talking about analysis or forecast ? Please detail both aspects regarding MLD.

p.11 l.5: I totally agree with the author: an additional experiment using FOAM with a

6h window or CPLDA with a 12h window is needed to disentangle this effect from other possible factors such as atmospheric forcings, turbulent fluxes schemes, . . . Hence, I strongly suggest to the author to do this simulation if the modifications of the systems are not too heavy and depending on available computing resources. I would greatly improve the manuscript discussion and strengthen the results presented here.

p.11 section 3.3: please add a word if the product is also assimilated or other products related to oceanic currents. Please make a better distinction between analysis and forecast biases and RMSE.

p.11 l.20: the sentence is in contradiction with what is stated above at p.11 l.15-16 ("bias and RMSE stable during forecast"). Please correct this or give more explanations.

p. 12 l.1-2: is it possible to confirm this statement by comparing directly the number of assimilated observations between both systems please ? Would it be possible to conduct the same kind of tests to address other questions or comments in the manuscript related to the difference in term of assimilation time window or observation number ?

p.12 l.6-7: again, an additional test with a different time window would strongly clarify these statements and give a real matter for discussion.

p.12 l.10-11: again, an additional 1-month experiment using the new MDT would clarify this.

p.12 fig. 5:Figure 5 is not described nor employed in the text. Please supress it or comment it.

p.13 l.2-3: on the contrary, it has been shown that coupled models have more EKE damping than ocean-only forced models (see Renault et al. 2016 for example). Consequently, this explanation is incomplete or erroneous.

p.14-15 – section 3.4:

There is no numbers in this section text to quantify the differences between atmospheric forcings heat fluxes and stresses. It will make the comparison easier and more "physical" if you add them.

p.15 l.1-3: the shortwave (SW) bias pattern suggest an SW overestimation on all eastern boundary upwelling systems. This is usually related to an underestimation of the low-level stratiform clouds which atmospheric models have difficulties to represent. A comment about this atmospheric bias should be added if it is also present in NWP.

p.15 l.21: please give a quantification of the contributions of atmospheric state and latent heat to the total net heat flux difference.

p. 15 – wind stress analysis: I have some difficulties to correctly understand this part. First, there is no distinction between atmospheric analysis and forecasts. I suppose analysis are relatively close to MetOp observations as they are probably assimilated, while 7-days forecasts must have quickly increasing surface stress errors. Consequently, it is difficult to guess where the stress differences between CPLDA and MetOp come from. Then, CPLDA wind stress is strongly and globally underestimated ($\sim$-0.1 N/m2) compared to MetOp. This bias is far larger than the difference between CPLDA and FOAM ($\sim$+0.5 N/m2). I don't think this is related to bulk formulae differences (COARE 3.0 is nearly identical to COARE 3.5 except in very strong wind conditions which statistically almost never occur). It can be partially related to the fact that scaterometers usually provide surface stress in atmospheric neutral stability condition, but it cannot explain the global underestimation observed here. This can also be related to the absolute or relative wind/current coupling in CPLDA. But finally, I don't understand why this global stress underestimation is not associated with warm SST and shallow MLD biases in both FOAM and CPLDA as it should be the case. Please explain clearly why it is not the case.

p.16 – conclusion: Conclusion should be updated accordingly to the modifications done in the manuscript.

---

## Referee Comment (RC2) · Anonymous Referee #2 · 15 Mar 2019

General Comments:

This paper presents an evaluation of the ocean component of an atmosphere-ocean coupled prediction system that has been running in real-time at the UK Metoffice and builds upon the work presented by Lea et al. (2015). In particular, this system employs a "weakly coupled" data assimilation approach, which affects the analysis and forecast quality in a number of ways. A systematic approach is used to provide an objective evaluation of the different state variables (sea surface temperature, mixed layer depth, currents) against accepted reference datasets, including other operational forecast products. The presentation is generally clear, but remains fairly descriptive,

limiting the conclusions that can be drawn from the study. Many of the differences between the coupled and operational forecasting systems are due to differences in particular parameter choices and not directly due to the coupled model or "weakly coupled data assimilation" themselves. As a result, the value of the paper is limited to a presentation of the baseline skill of the system. Even this is put somewhat in question, as the results (for currents and sea level) are shown to be sensitive to differences in data assimilated in the two systems resulting in much larger errors in the coupled system. The paper would benefit significantly from having additional experiments to clarify the impact of the different parameter choices and a more in-depth analysis of results. Additionally, the paper would benefit from some reorganization as the differences in fluxes are described near the end of the paper, much of which is needed to understand the differences being presented in previous sections.

I recommend the paper for publication but strongly encourage the authors to deepen the analysis of the results and provide additional experiments to shed-light on the source of differences.

Specific Comments:

Section 1: The introduction is limited to activities at the UK Metoffice and would benefit greatly from some context into why weakly coupled data assimilation is of interest and previous efforts. For example, a study has recently been published that should be referenced in this discussion (Browne et al., 2019). Moreover, several centers are now running coupled forecasting systems operationally, which should be mentioned to provide context into UK Metoffice efforts. Browne, P.A., de Rosnay, P., Zuo, H., Bennett, A. and Dawson, A., 2019. Weakly Coupled Ocean–Atmosphere Data Assimilation in the ECMWF NWP System. Remote Sensing, 11(3), p.234.

Pg1, L19: Johns et al. (2012) is a technical report that appears not to be publically available. There are a number of such reports referenced here. If these reports are only available upon request, than greatly detail regarding their content should be included

in the text. Also for Lea et al. (2013).

Pg2, L14: "To avoid duplication of previous work . . .we did not investigate the impact on the diurnal cycle already covered in Lea et al., (2013)". This is a fairly important point that is referred to later in terms of errors in temperature and mixed layer depth. Since Lea et al., (2013) is only a technical report these results could be included here (or at least explained more clearly).

Pg. 3, L1: COARE3.0, please add a reference. Also this is not noted in Table 1.

Pg. 3, L9: Lea et al. (2015) note significant difficulties associated with runoff. How is runoff handled here? Are there differences between FOAM and CPLDA?

Pg. 3, L32: "Haney retroaction". Please provide a reference and more details. I.e. what is the timescale of the relaxation? To what fields?

Pg.4, L19: It says here that CPLDA uses a fixed freezing temperature, which differs from what is indicated in Table1.

Pg. 5, L27-32: It may be useful to note here that there is also an impact on the observation cutoff time that affects the number of SLA observations used.

Pg. 6, Sec2.6: If the CPLDA system has been running operationally since October 2016, why is it necessary to run additional experiments? If the aim here is to benchmark the skill of CPLDA as compared to FOAM why not use the operational runs? If it possible to make additional runs, could a longer run with the SLA observation cutoff corrected be produced? This would make the evaluation much more straightforward.

Pg. 6, L17: "There are a few differences. . .". What is the impact of these differences? If they are not too significant an analysis of the operational run compared to FOAM for Oct. 2016 onwards could add additional insight to the paper.

Pg. 7, L7: SLA is an important variable providing information regarding the circulation and mesoscale activity. It is also relevant here to provide context to the errors in 15m

velocity. I would strongly encourage you to include these results, despite the poor results. Ideally, an additional run could be made with the cutoff issue corrected.

Pg. 7, L16: It is argued that sea ice is not evaluated because of a difference in freezing temperature. However, there are differences in many other aspects as well (bulk formulae, SLA observation cutoff, assimilation window). If the aim is to benchmark the performance of the system, than an evaluation of the sea ice component should be included as well.

Pg. 7, L20: Is it the instantaneous SST or a daily average?

Pg. 8, L9: Should this be 0.01K? The global mean difference in bias between CPLDA and PSY4 appears to be less than 0.05K. Also, if OSTIA is cooler than CPLDA and PSY4 assimilates OSTIA, why is PSY4 warmer than CPLDA?

Pg. 8, L19: The Fiedler reference is only "in prep". I recommend providing a more thorough explanation with a "personal communication" to Fiedler.

Pg. 9, Fig. 2d: What is the global mean value of this? It would be helpful to include this on all four panels. It appears to me that it is positive, which seems in conflict with Fig. 3b that shows CPLDA has a more negative global near-surface temperature increment.

Pg.10, L5-14: This asymmetrical effect of SST increments is quite an important difference between CPLDA and FOAM and warrants further explanation. Perhaps an example figure showing profiles and how the increments deteriorate the profile. However, the argument is based on the fact that large increments in CPLDA then create a large asymmetrical effect on SST. But why are the increments larger in the first place?

Pg. 10, Fig. 3: What is the explanation for the large increments between 50-100m and associated increase in RMSE in CPLDA? Is it related to the asymmetric effect of SST increments? If so, this should be described.

Pg. 10, L16: How is the mixed layer depth calculated? The Fig. 4 caption quotes Kara (2000), but some explanation of the methodology used would be appropriate.

Pg. 10, L17: "mean error .. is -5.2m". The text (Pg. 10, line2) notes that CPLDA has a deeper mixed layer than FOAM, shouldn't this value be positive?

Pg. 11, Fig. 4b: Please state whether positive values are deeper or shallower in CPLDA.

Pg. 11, L2-4: It is difficult to follow this reasoning when we haven't yet seen the differences in wind stress. This section would be easier to follow if Sec3.4 was presented first providing context regarding differences surface fluxes.

Pg. 11, L7: "Predictions of ocean current are important for marine activities". It would be helpful to expand somewhat since this statement is vague. Also, it should be "currents". Pg. 11, L19: Are differences in tropical Pacific along capable of affecting global statistics this much? Or are there large differences elsewhere as well?

Pg. 11, L21: If the shorter window affects the increments in a direct manner that influences the results to this degree, I think it warrants some further description. Perhaps some examples of increments in this area. This is extremely relevant to the overall effort to produce a "weakly coupled" data assimilation system and should be elaborated upon.

Pg. 11, L21: By how much are the number of observations limited? This too could be expanded upon to clarify the impact of the assimilation window on results.

Pg. 12, L3: If the operational system uses the updated scheduling than why not evaluate it instead of the experiment with the reduced number of observations?

Pg.12, L7: If the unrealistic currents are not always caused by the reduced number of SLA observations, than perhaps the observations are simply covering up an existing issue in the coupled model. This issue warrants further investigation, it its central to many of the results presented.

Pg. 14, L8-9: A "reduced heat loss" should lead to warmer SSTs shoudn't they?

Pg. 14, L9-10: While the differences in short wave radiation and latent heat clearly contribute to differences in net heat flux difference. These fields both have large scale patterns, whereas there is a notable small scale pattern in the net heat flux difference not explained.

Pg. 17, L25-27: As noted above, the argument about the noisier increments is quite important to the overall conclusions. I would recommend a deeper analysis be made of this issue.

Pg. 19, L8: should be "well-placed"
* * *

---

## Author Comment (AC1) · 12 Apr 2019

We are grateful to Reviewer1 for the time and efforts assessing our work, and for providing comments. These comments are reproduced below, with our responses indicated with asterisks.

General comments: This study presents a new ocean-atmosphere coupled operational system (CPLDA) with weakly coupled data assimilation developed at Met Office. In the first part, the model components are described and compared to previous coupled and uncoupled systems (GloSea and FOAM respectively). The weakly coupled data assimilation method from Lea et al. (2013, 2015) is also detailed. This system is operated

[Figure]

and evaluated during one year (2015) with 6-hour analysis and daily 7-day forecast. The system performances are then compared with the operational ocean-only FOAM (Ryan et al. 2015) and Mercator PSY4 systems and with different observational (SST, MLD, 15 m velocities, SLA) datasets. Atmospheric forcings and turbulent fluxes are also analysed. In summary, the CPLDA system performs as well as ocean-only systems despite its increased complexity and constitute a promising first step toward a fully coupled operational system. The manuscript provides a precise description of the CPLDA system, underlining the benefits and limitations compared to other MetOffice systems and observations. It is globally well written and scientific analyses are seriously presented. However, I recommend the following modifications to improve the manuscript quality and understanding. I have the feeling that some sensitivity tests and analysis are still missing to better cover the system description and validation. This is mainly due to the fact that besides the ocean model similar configurations between CPLDA and FOAM, there a large number of differences between the two systems. Consequently, it is sometimes difficult for the authors to assess precisely the differences between the systems because too much parameters change at the same time. Some additional sensitivity tests allow to better understand those differences (for example, the one month experiment is a different scheduling for the DA). But others sensitivity tests are missing to clearly understand and to disentangle the numerous modifications between the systems. As a consequence, some analysis are not very convincing because of this lack of sensitivity experiments. Depending on the authors will and capacity to run these additional simulations, some corrections will have to be made to the text to confirm or infirm the hypothesis proposed. If no supplementary experiments can be done, some comments must be added to the text to emphasize the limitations of the study. Because atmospheric forcings and surface fluxes are strongly related to SST, upper temperature and MLD, the section 3.4 should be moved before the section 3.3 about velocities (that I would rename "upper or ML velocities"). It is not always clear if the authors are talking about the analysis, the forecast or both in the whole manuscript. It makes the study understanding less clear to follow and more

difficult to understand. I think the manuscript can be easily improved by clearly stating this.

\*\*Experiments are currently being undertaken using an ocean-only model to look at the impact of reducing the assimilation window from 24 to 6 hours. These experiments should decompose the effect of the shortened assimilation window from the effect of the coupling and aid in the interpretation of the comparisons to the FOAM system in the manuscript. It is important to note that the scheduling of the operational CPLDA system and the availability of observations will not be investigated in these experiments. It is hoped that the results from these experiments will be available before the final re-submission of the manuscript and the text edited accordingly, but expect this would only be possible if an additional month was provided to allow time for this to be done.

We agree with reviewer1 about the organisation of the paper and we will move the section on surface fluxes before the velocities.\*\*

Specific comments: p.1 - Introduction: The introduction is not really relevant because it is mostly a repeat of the system description in section 2. The introduction should be improved to better describe the context of this study, i.e to describe the main physical and technical arguments in favour of developing a coupled system with weakly coupled assimilation (instead of independent systems and assimilations). A description of what is done in other operational centres (ECMWF, NCEP, . . .) could also help to better understand the framework and the interest of the system presented here.

\*\*To provide more context the Met Office efforts, we will update the introduction and in particular we propose to add:

" Coupled systems are used in wide range of applications (short and medium range forecasts, seasonal forecasts, climate prediction and future scenario projections) and improving the initialization of these systems can play a significant part to reduce the development of errors. Using separate atmosphere and ocean analysis to initialize a coupled system can result in an imbalanced system. The imbalance can cause

an initialisation shock that could potentially increase the development of errors during the forecast. Using a coupled data assimilation approach has been shown to reduce this initialisation shock (Mulholland, 2015). In recent years, numerous research centres have developed operational coupled data assimilation systems. For instance, the European Centre for Medium-Range Weather Forecast (ECMWF) has developed a weakly coupled ocean-atmosphere data assimilation system (Browne, 2019). Their results show that using their coupled data assimilation system reduces forecast errors compared with forecast initialised from uncoupled analysis. The National Centers for Environmental Prediction (NCEP) is using a coupled data assimilation system for seasonal and sub seasonal scale predictions (Saha et al, 2014). Penny et al (2017) proposes an overview of the efforts made on coupled data assimilation in operational centres. It shows the diversity of the approaches available. At JAMSTEC, they developed a low resolution strongly coupled system used for experimental seasonal and decadal prediction while NRL coupled model is initialised by separate analysis but uses a high-resolution ocean component (1/25)." **

p.3 l.10-12: it is not clear if VarBC is activated or not in the present study, please specify this in the text.

**We propose to add: "In the present paper, all the CPLDA experiments use VarBC." **

p.5 l.30: please justify why such changes regarding IAU and observation operator were made in CPLDA compared to FOAM.

**3-hour IAU are used so for the 00Z analyses the full increments will have been added to the background. These analyses can subsequently be used by groups running initialised coupled forecasts.**

p.6 l.20: scheduling of the operational system ? **it will be corrected**

p.6 l.25: I don't understand why some results should come from the operational version of the system and others from "test" version ? I find this confusing. Perhaps it should

be better to talk about the actual operational version only in the conclusion to avoid any confusion.

**We propose to include a paragraph on the SLA to the paper. The results from the test experiment 2015 highlighted the degradation of the SLA statistics compared to FOAM cause by the scheduling. The results from the operational system with updated scheduling are presented only to highlight how this problem was solved.**

p.6 l.31: please add Lellouche et. al 2018 as a reference for Mercator-Ocean PSY4 system.

**This reference will be added : Lellouche, Jean-Michel & Greiner, Eric & Le Galloudec, Olivier & Garric, Gilles & Regnier, Charly & Drevillon, Marie & Mounir, Benkiran & Testut, Charles-Emmanuel & Bourdalle-Badie, Romain & Gasparin, Florent & Hernandez, Olga & Levier, Bruno & Drillet, Yann & Remy, Elisabeth & Traon, Pierre-Yves. (2018). Recent updates to the Copernicus Marine Service global ocean monitoring and forecasting real-time 1/12° high-resolution system. Ocean Science. 14. 1093-1126. 10.5194/os-14-1093-2018.**

p7 l.9-11: I think these results are important and should be added to the paper as a distinct section or paragraph with a dedicated figure to illustrate the differences in terms of SLA bias and RMSE between FOAM and the 2 versions of CPLDA with the different schedulings. It is also unclear if the results presented here are those from the current operational system or from the version used in the study ("old scheduling").

**We agree with the comment and will add a figure to illustrate these results and make clear that the version used for this assessment uses the old scheduling.

We propose to add:

"CPLDA 2015 and FOAM sea surface height are assessed against observations using class4 statistics (Ryan et al., 2015). The observations used are provided by CMEMS and include data from Altika, Cryostat and Jason-2 satellites. Altimeter bias correction

is applied to the observations. For each model comparison against the satellite observations, it is important to use the model own altimeter bias. The altimeter bias contains information from the model mesoscale so correcting observations using the altimeter bias from one model to assess a second model penalised this second model.

Figure 1 shows a timeseries the sea level anomaly (SLA) difference statistics assessed against CMEMS observations. In the 2015 experiment with the old scheduling, CPLDA SLA root-squared-mean error (RMSE) is significantly larger than FOAM SLA RMSE (Figure 1a). The larger RMSE in CPLDA can be attributed to the difference in the number of SLA observations assimilated by both systems. Figure 2 shows the number of observations assimilated by both systems in 2015 the number assimilated by CPLDA is significantly smaller than the number assimilated by FOAM. Differences in scheduling can explain this. In comparison with FOAM, the CPLDA 2015 experiment "best analysis" runs earlier in the day so fewer observations are available. Following these results, the scheduling of the CPLDA operational system was updated in April 2018 in order to improve CPLDA performance. The best analyses at 0600Z, 1200Z and 1800Z are now delayed allowing more observations to be assimilated. This change along with a change in the Met Office database (MetDB) that now allows a more frequent ingestion of SLA observations, has resulted in a significant reduction of the CPLDA RMSE (see figure 1b).

Figure 1: SLA class4 statistics with respect to CMEMS satellite product (Jason2, Cryosat and Altika) for CPLDA and FOAM. a. For the long CPLDA experiment in 2015 with the old scheduling. b. from the operational systems for 2018, the CPLDA scheduling was updated to allow more observations to be assimilated in April 2018.

Figure 2: Number of SLA observations assimilated in FOAM and CPLDA in the 2015 experiment" **

p.14 section 3.4 should be moved close to the sections 3.1 and 3.2 as it is directly related to changes in SST and MLD. It would improve the readability of the manuscript.

**We agree with this comment and will change the order of the sections accordingly.**

p.7 l.28-30 This hypothesis should be directly tested by doing some additional sensitivity experiments using a 24h window with CPLDA or a 6h window with FOAM to confirm it or not.

**Experiments are currently being undertaken using an ocean-only model to look at the impact of reducing the assimilation window from 24 to 6 hours. The results from that experiment will be added to the manuscript.**

p.7 l.30 "in CPLDA" -> "in CPLDA analysis": it is not clear when the author is talking about the analysis or the forecast. Please specify it everywhere it is necessary in the text.

**We will update the text accordingly**

p.7 l.31: is there a way to measure this "overfitting" ? It is not clear why FOAM SST forecast performs better than CPLDA and how is it related to this overfitting.

**Overfitting can be measured by withholding a sub-set of observations from the DA, these are then used for validation. This approach is not pragmatic in an operational setting where you want to get the best analysis possible at runtime. Assessing the overfitting will be undertaken in the assimilation window experiments detailed above. These experiments will include investigating scaling the background errors for the 6-hour window (current values calculated from system using 24 hour window). It is thought that in the current CPLDA system background errors are too large (less weight in the DA) thus observations are given too much weight in the assimilation and introducing observation noise.

Similar performance is seen in CPLDA and FOAM forecasts at lead times of 36 hours and beyond.**

p.8 fig.1: the forecast duration is 7 days (168h), please expand the figure axis accordingly

*Addition of an extra day to this figure would require a significant amount of extra data processing which we do not feel would add much

p.8 l.19: it could be interesting to give the effective SST resolution for both system even without showing the figure to get an idea of the scale range resolved.

**The effective SST resolution is the same for both system around 50km.**

p.9 section 3.2: please add PSY4 in the analysis and Figure 3 if possible to be more coherent with others sections and to get a more complete comparison between products.

**Although this would be a useful addition to the manuscript it would require a substantial amount of data processing. As the scope of the paper is the CPLDA system the comparison to FOAM is valuable as the systems are relatively similar which isn't the case for PSY4. **

p.9 l.9: please give the reference associated with this product.

**We will add the reference to ARGO product: IN-SITU_GLO_NRT_OBSERVATIONS_013_030 **

p.9 l.11: please give the range of the RMSE increase to be able to compare it to the SST RMSE. T rmse increased from 0.67 at day1 to 0.73 at day5

p.10 l.15: the warm bias and associated significant RMSE (_1_C) located at _100m is not discussed nor explained in the text. Please add a paragraph about this. If the discussion about King et al. 2018 results explain it, it should appear more explicitly in the text.

**This section has been re-written (see below) and the reviewer comments on RMSE addressed.

[revised manuscript text omitted]

p.10 l.17: are you talking about analysis or forecast ? Please detail both aspects regarding MLD.

**We will update the text to make it clear that the MLD results presented are from the analysis. During the forecast the rmse and bias persist (not shown).**

p.11 l.5: I totally agree with the author: an additional experiment using FOAM with a 6h window or CPLDA with a 12h window is needed to disentangle this effect from other possible factors such as atmospheric forcings, turbulent fluxes schemes, . . . Hence, I strongly suggest to the author to do this simulation if the modifications of the systems are not too heavy and depending on available computing resources. I would greatly improve the manuscript discussion and strengthen the results presented here.

\*\*An experiment to test the impact of the time window is ongoing see 1st comment.\*\*

p.11 section 3.3: please add a word if the product is also assimilated or other products related to oceanic currents. Please make a better distinction between analysis and forecast biases and RMSE.

\*\*The currents are not assimilated. We will update the text to make clear what is analysis what is forecast\*\*

p.11 l.20: the sentence is in contradiction with what is stated above at p.11 l.15-16 ("bias and RMSE stable during forecast"). Please correct this or give more explanations.

\*\*The bias and RMSE are stable during the forecast on a global scale but in the tropical pacific where there are large erroneous currents in the analysis the RMSE decreases during the forecast. \*\*

p. 12 l.1-2: is it possible to confirm this statement by comparing directly the number of assimilated observations between both systems please ? Would it be possible to conduct the same kind of tests to address other questions or comments in the manuscript related to the difference in term of assimilation time window or observation number ?

\*\*In the new paragraph on SLA we will add a plot highlighting the difference in the numbers of observation between FOAM and CPLDA 2015 experiments. We will address the question of the assimilation time window by running an experiment with FOAM and a 6-hour window. We don't have the time and resources to run more impact on the number of observations.\*\*

p.12 l.6-7: again, an additional test with a different time window would strongly clarify these statements and give a real matter for discussion.

\*\*Yes as above\*\*

p.12 l.10-11: again, an additional 1-month experiment using the new MDT would clarify

this.

\*\*The MDT will not be updated in the current iteration of the CPLDA system so these experiments will not be undertaken. The next generation O-A coupled system which is currently being developed uses an updated MDT from CNES-CLS09 to CNES-CLS13 and initial results look promising but these results are outside the scope of this paper. \*\*

p.12 fig. 5:Figure 5 is not described nor employed in the text. Please supress it or comment it.

\*\*We will remove Figure 5\*\*

p.13 l.2-3: on the contrary, it has been shown that coupled models have more EKE damping than ocean-only forced models (see Renault et al. 2016 for example). Consequently, this explanation is incomplete or erroneous.

\*\*We agree with the reviewer; the text is not clear. We propose to update it: "Eddy Kinetic Energy (EKE) was also compared in CPLDA, FOAM and PSY4. In an ocean only model, using the ocean velocity to calculate the winds tress has a damping effect on the eddies (Duhaut and Straub, 2006; Dawe and Thompson, 2006; Renault et al., 2016). Coupled system should not suffer from the same eddy damping. Despite a reduced damping effect in coupled system, we did not observe a higher EKE in CPLDA than in FOAM. At a 1/4 resolution, the mesoscale is poorly represented and most of the EKE is injected into the model by the SLA assimilation. Hence any differences in EKE between CPLDA and FOAM are mainly caused by differences in SLA observations assimilated, or impacts of the shorter assimilation window, rather than due to a reduced eddy damping by the wind-stress." \*\*

p.14-15 – section 3.4: There is no numbers in this section text to quantify the differences between atmospheric forcings heat fluxes and stresses. It will make the comparison easier and more "physical" if you add them.

\*\*We calculated the average value for the flux differences plotted in Figure 8. On average, CPLDA loose more heat than FOAM which contributes to the colder SST observed in CPLDA. If the differences in short wave radiation show some strong patterns with the largest difference in the eastern boundary upwelling on average its contribution is limited. On average for the global ocean, the strongest contributions come from the latent heat differences.

Average for total heat flux diff (CPLDA-FOAM) = -3.9 W/m2 Average for latent heat diff (CPLDA – CPLDA with CORE)=+9.61 W/m2 (1) Average for latent heat diff (CPLDA – NWP)=-14.50 W/m2 (2) Average for short wave diff (CPLDA-FOAM) =+1.05 W/m2 (3) (1)+(2)+(3) = -3.84 W/m2 \*\*

p.15 l.1-3: the shortwave (SW) bias pattern suggest an SW overestimation on all eastern boundary upwelling systems. This is usually related to an underestimation of the low-level stratiform clouds which atmospheric models have difficulties to represent. A comment about this atmospheric bias should be added if it is also present in NWP.

\*\*The stratocumulus clouds are under estimated in Met office atmosphere model and causes a short-wave overestimation on the eastern boundary upwelling systems. The bias gradually reduces as the resolution increases as the resolution is lower in the atmosphere component of CPLDA (40km) than in the NWP fluxes forcing FOAM (17km) the overestimation is larger in CPLDA.\*\*

p.15 l.21: please give a quantification of the contributions of atmospheric state and latent heat to the total net heat flux difference.

\*\*Globally, the differences in latent heat due to the different bulk formulae (+9.61 W/m2) and the different atmosphere (-14.50 W/m2) are the main contributions to the difference in heat fluxes.\*\*

p. 15 – wind stress analysis: I have some difficulties to correctly understand this part. First, there is no distinction between atmospheric analysis and forecasts. I suppose

analysis are relatively close to MetOp observations as they are probably assimilated, while 7-days forecasts must have quickly increasing surface stress errors. Consequently, it is difficult to guess where the stress differences between CPLDA and MetOp come from. Then, CPLDA wind stress is strongly and globally underestimated (_-0.1 N/m2) compared to MetOp. This bias is far larger than the difference between CPLDA and FOAM (_+0.5 N/m2). I don't think this is related to bulk formulae differences (COARE 3.0 is nearly identical to COARE 3.5 except in very strong wind conditions which statistically almost never occur). It can be partially related to the fact that scaterometers usually provide surface stress in atmospheric neutral stability condition, but it cannot explain the global underestimation observed here. This can also be related to the absolute or relative wind/current coupling in CPLDA. But finally, I don't understand why this global stress underestimation is not associated with warm SST and shallow MLD biases in both FOAM and CPLDA as it should be the case. Please explain clearly why it is not the case.

**All wind stress presented are from the analysis. We are currently investigating why the difference are larger than expected and we will add the results from that investigation to the manuscript.

The negative increment suggests the model is too warm, the MLD is constrained by the assimilation of not only profile observations but also the SST observations as described in section 3.2. Without a free-run it's hard to know what the MLD would look like due to wind-stress differences only without the effect of the surface increments on the MLD. **

p.16 – conclusion: Conclusion should be updated accordingly to the modifications done in the manuscript.

**We will update the conclusion**
* * *
[Figure]

[Figure]

**Fig. 1.** SLA class4 statistics with respect to CMEMS satellite product (Jason2, Cryosat and Al-tika) for CPLDA and FOAM. a. For the long CPLDA experiment in 2015 with the old scheduling. b. from the operational

[Figure]

**Fig. 2.** Number of SLA observations assimilated in FOAM and CPLDA in the 2015 experiment

---

## Author Comment (AC2) · 12 Apr 2019

We are grateful to Reviewer2 for the time and efforts assessing our work, and for providing comments. These comments are reproduced below, with our responses indicated with asterisks.

General Comments: This paper presents an evaluation of the ocean component of an atmosphere-ocean coupled prediction system that has been running in real-time at the UK Metoffice and builds upon the work presented by Lea et al. (2015). In particular, this system employs a "weakly coupled" data assimilation approach, which affects the analysis and forecast quality in a number of ways. A systematic approach is used to

provide an objective evaluation of the different state variables (sea surface temperature, mixed layer depth, currents) against accepted reference datasets, including other operational forecast products. The presentation is generally clear, but remains fairly descriptive, limiting the conclusions that can be drawn from the study. Many of the differences between the coupled and operational forecasting systems are due to differences in particular parameter choices and not directly due to the coupled model or "weakly coupled data assimilation" themselves. As a result, the value of the paper is limited to a presentation of the baseline skill of the system. Even this is put somewhat in question, as the results (for currents and sea level) are shown to be sensitive to differences in data assimilated in the two systems resulting in much larger errors in the coupled system. The paper would benefit significantly from having additional experiments to clarify the impact of the different parameter choices and a more in-depth analysis of results. Additionally, the paper would benefit from some reorganization as the differences in fluxes are described near the end of the paper, much of which is needed to understand the differences being presented in previous sections. I recommend the paper for publication but strongly encourage the authors to deepen the analysis of the results and provide additional experiments to shed-light on the source of differences.

\*\*Experiments are currently being undertaken using an ocean-only model to look at the impact of reducing the assimilation window from 24 to 6 hours. These experiments should decompose the effect of the shortened assimilation window from the effect of the coupling and aid in the interpretation of the comparisons to the FOAM system in the manuscript. It is important to note that the scheduling of the operational CPLDA system and the availability of observations will not be investigated in these experiments. It is hoped that the results from these experiments will be available before the final resubmission of the manuscript and the text edited accordingly, but expect this would only be possible if an additional month was provided to allow time for this to be done.

We agree with reviewer2 about the organisation of the paper and we will move the

section on surface fluxes before the velocities. We will also add a paragraph on the SLA results.**

Specific Comments:

Section 1: The introduction is limited to activities at the UK Metoffice and would benefit greatly from some context into why weakly coupled data assimilation is of interest and previous efforts. For example, a study has recently been published that should be referenced in this discussion (Browne et al., 2019). Moreover, several centers are now running coupled forecasting systems operationally, which should be mentioned to provide context into UK Metoffice efforts. Browne, P.A., de Rosnay, P., Zuo, H., Bennett, A. and Dawson, A., 2019. Weakly Coupled Ocean–Atmosphere Data Assimilation in the ECMWF NWP System. Remote Sensing, 11(3), p.234.

**To provide more context the Met Office efforts, we will update the introduction and in particular we propose to add:

" Coupled systems are used in wide range of applications (short and medium range forecasts, seasonal forecasts, climate prediction and future scenario projections) and improving the initialization of these systems can play a significant part to reduce the development of errors. Using separate atmosphere and ocean analysis to initialize a coupled system can result in an imbalanced system. The imbalance can cause an initialisation shock that could potentially increase the development of errors during the forecast. Using a coupled data assimilation approach has been shown to reduce this initialisation shock (Mulholland, 2015). In the recent years, numerous research centres have developed operational coupled data assimilation systems. For instance, the European Centre for Medium-Range Weather Forecast (ECMWF) has developed a weakly coupled ocean-atmosphere data assimilation system (Browne, 2019). Their results show that using their coupled data assimilation system reduces forecast errors compared with forecast initialised from uncoupled analysis. The National Centers for Environmental Prediction (NCEP) is using a coupled data assimilation system for

seasonal and sub seasonal scale predictions (Saha et al, 2014). Penny et al (2017) proposes an overview of the efforts made on coupled data assimilation in operational centres. It shows the diversity of the approaches available. At JAMSTEC, they developed a low resolution strongly coupled system used for experimental seasonal and decadal prediction while NRL coupled model is initialised by separate analysis but uses a high-resolution ocean component (1/25). " **

Pg1, L19: Johns et al. (2012) is a technical report that appears not to be publically available. There are a number of such reports referenced here. If these reports are only available upon request, than greatly detail regarding their content should be included in the text. Also for Lea et al. (2013).

**We propose to add details for both references:

"In Johns et al (2012), a large set of global coupled atmosphere-ocean-sea ice 15-day hindcasts were initialised from separate atmosphere and ocean-ice analysis. Compared to uncoupled forecast, the coupled 15-day hindcasts show improved forecasting skills especially in the Tropics, region strongly coupled where small changes in SST can exert a major influence on patterns of convection and remote responses to diabatic heating. Both atmosphere and ocean biases benchmarked against control simulations suggest a more vigorous water cycle, improved large-scale circulation, enhanced convection, stronger teleconnections and somewhat improved representation of the Madden-Julian oscillations. They conclude that significant benefits should arise from full integration of coupled NWP system."

"Lea et al (2013) developed a coupled data assimilation (DA) system with the aim of improving the initialisation of coupled forecasts for various time ranges from short-range out to seasonal. They used a "weakly" coupled data assimilation approach whereby the coupled model is used to provide background information for separate ocean/sea-ice and atmosphere/land analyses. The increments generated from these separate analyses are then added back into the coupled model. They investigate the impact

of weakly coupled DA on analysis and forecast skills up to 5 days. They showed that ocean and atmospheric analyses produced using coupled DA to be very similar to the analyses produced using separate ocean/atmosphere data assimilation" **

Pg2, L14: "To avoid duplication of previous work . . .we did not investigate the impact on the diurnal cycle already covered in Lea et al., (2013)". This is a fairly important point that is referred to later in terms of errors in temperature and mixed layer depth. Since Lea et al., (2013) is only a technical report these results could be included here (or at least explained more clearly).

**We propose to add: "They investigated the impact of the initialisation on the diurnal cycle of SST has been investigated by comparing to geostationary satellite data from the SEVIRI. They looked at the diurnal SST range and showed that the initialisation of the coupled forecast has very little impact on the diurnal range. Both Coupled and uncoupled forecasts appear to represent the regions of larger diurnal cycle observed by the satellite well, although they perhaps under-estimate the range in the central North Atlantic " **

Pg. 3, L1: COARE3.0, please add a reference. Also this is not noted in Table 1.

**This reference will be added Fairall, C.W., E.F. Bradley, J.E. Hare, A.A. Grachev, and J.B. Edson, 2003: Bulk Parameterization of Air–Sea Fluxes: Updates and Verification for the COARE Algorithm. J. Climate, 16, 571–591, https://doi.org/10.1175/1520-0442(2003)016<0571:BPOASF>2.0.CO;2

We will update table 1 accordingly. **

Pg. 3, L9: Lea et al. (2015) note significant difficulties associated with runoff. How is runoff handled here? Are there differences between FOAM and CPLDA?

**Due to the issues in Lea et al. (2015), we are for the moment using the same climatological run off as in FOAM. Since Lea et al. (2015), runoff from the atmosphere/land component may well have improved and plan to revisit using this in future.**

Pg. 3, L32: "Haney retroaction". Please provide a reference and more details. I.e. what is the timescale of the relaxation? To what fields?

\*\*A Haney flux correction (Haney, 1971) is applied to the sea surface salinity (SSS) based on the difference between the model and climatology. The magnitude of the restoring on SSS is -33.33 mm/day/psu.

Haney,R. 1971. Surface boundary conditions for ocean general circulation models. J. Phys. Oceanogr. 1: 241–248. \*\*

Pg.4, L19: It says here that CPLDA uses a fixed freezing temperature, which differs from what is indicated in Table1.

\*\*Reviewer is correct the table 1 will be corrected.\*\*

Pg. 5, L27-32: It may be useful to note here that there is also an impact on the observation cutoff time that affects the number of SLA observations used.

\*\*We propose to add this: "It is important to note that difference in observation cutoff due to difference in operational scheduling impact the number of observations used in each system." \*\*

Pg. 6, Sec2.6: If the CPLDA system has been running operationally since October 2016, why is it necessary to run additional experiments? If the aim here is to benchmark the skill of CPLDA as compared to FOAM why not use the operational runs? If iit possible to make additional runs, could a longer run with the SLA observation cutoff corrected be produced? This would make the evaluation much more straightforward.

\*\*Since October 2016, the operational CPLDA suite had several updates and bug fixes so we could not have a long enough period with consistent settings to be able to properly assess the system. We will not be able to produce a longer run with the corrected cutoff but results from the Operational CPLDA system with the updated scheduling allowing more SLA observations to be assimilated now shows similar SLA statistics than the FOAM system.\*\*

Pg. 6, L17: "There are a few differences. . .". What is the impact of these differences? If they are not too significant an analysis of the operational run compared to FOAM for Oct. 2016 onwards could add additional insight to the paper.

**There are differences between the operational system in 2016 and the same system in 2018 (several upgrades and bug fixes). The 2015 experiment was made to assess the 2018 version of the operational system. As noted in the manuscript some differences are unavoidable (different observations available) and the other main difference is the scheduling that was update in the operational system in June 2018 as a result of the assessment of the 2015 experiment. **

Pg. 7, L7: SLA is an important variable providing information regarding the circulation and mesoscale activity. It is also relevant here to provide context to the errors in 15m velocity. I would strongly encourage you to include these results, despite the poor results. Ideally, an additional run could be made with the cutoff issue corrected.

**The current CPLDA operational system with the updated scheduling allowing more observations to be assimilated has SLA statistics similar to FOAM. A section on the SLA results will be added to the manuscript.

"CPLDA 2015 and FOAM sea surface height are assessed against observations using class4 statistics (Ryan et al., 2015). The observations used are provided by CMEMS and include data from Altika, Cryostat and Jason-2 satellites. Altimeter bias correction is applied to the observations. For each model comparison against the satellite observations, it is important to use the model own altimeter bias. The altimeter bias contains information from the model mesoscale so correcting observations using the altimeter bias from one model to assess a second model penalised this second model. Figure 1 shows a timeseries the sea level anomaly (SLA) difference statistics assessed against CMEMS observations. In the 2015 experiment with the old scheduling, CPLDA SLA root-squared-mean error (RMSE) is significantly larger than FOAM SLA RMSE (Figure 1a). The larger RMSE in CPLDA can be attributed to the difference in the number

of SLA observations assimilated by both systems. Figure 2 shows the number of observations assimilated by both systems in 2015 the number assimilated by CPLDA is significantly smaller than the number assimilated by FOAM. Differences in scheduling can explain this. In comparison with FOAM, the CPLDA 2015 experiment "best analysis" runs earlier in the day so fewer observations are available. Following these results, the scheduling of the CPLDA operational system was updated in April 2018 in order to improve CPLDA performance. The best analyses at 0600Z, 1200Z and 1800Z are now delayed allowing more observations to be assimilated. This change along with a change in the Met Office database (MetDB) that now allows a more frequent ingestion of SLA observations, has resulted in a significant reduction of the CPLDA RMSE (see figure 1b).

Figure 1: SLA class4 statistics with respect to CMEMS satellite product (Jason2, Cryosat and Altika) for CPLDA and FOAM. a. For the long CPLDA experiment in 2015 with the old scheduling. b. from the operational systems for 2018, the CPLDA scheduling was updated to allow more observations to be assimilated in April 2018.

Figure 2: Number of SLA observations assimilated in FOAM and CPLDA in the 2015 experiment" **

Pg. 7, L16: It is argued that sea ice is not evaluated because of a difference in freezing temperature. However, there are differences in many other aspects as well (bulk formulae, SLA observation cutoff, assimilation window). If the aim is to benchmark the performance of the system, than an evaluation of the sea ice component should be included as well.

**We propose include a figure illustrating the sea ice results. "Sea ice:

Figure 3 shows the sea ice extent and volume respectively for CPLDA and FOAM best analyses. Figure a shows that the sea ice extent is similar between CPLDA and FOAM and is also comparable to OSTIA. Figure b shows that the sea ice volumes are similar in the Arctic but in the Antarctic CPLDA has a reduced volume compared to FOAM

in the through Southern Hemisphere winter and spring. Differences in volume can be attributed to differences in sea ice thickness probably caused by differences in freezing temperature treatment (see section 2.3 Sea Ice component). Figure 4 shows the sea ice extent forecast for the melting season in Arctic (August) and Antarctic (February) for CPLDA and FOAM. Both models have the tendency to melt too much ice during the forecast especially in Arctic, in the Antarctic the exaggerated melting in CPLDA is slightly reduced in FOAM."

Figure 3: Time series of Sea Ice extent (left) and sea ice volume (right) for CPLDA (2015 experiment) and FOAM

Figure 4: Sea ice extent forecast for CPLDA (2015 experiment) and FOAM during the melting season (February in Antarctic and August in Arctic) **

Pg. 7, L20: Is it the instantaneous SST or a daily average?

**We use the daily average it will be clarify in the text.**

Pg. 8, L9: Should this be 0.01K? The global mean difference in bias between CPLDA and PSY4 appears to be less than 0.05K. Also, if OSTIA is cooler than CPLDA and PSY4 assimilates OSTIA, why is PSY4 warmer than CPLDA?

**Global average difference between CPLDA and OSTIA is 0.06K the text has been updated. Using Class4 statistics from the GODAE model inter-comparison (see figure 5 - not for inclusion in paper) PSY4 (purple) is generally biased warm and OSTIA(grey) is generally biased cold relative to surface drifters, this is consistent with the results presented for 2015. This reasons for the difference in SST characteristics between the systems despite PSY4 assimilating OSTIA could be due to model biases, atmospheric forcing etc but diagnosing these is outside the scope of this paper.

Figure 5. Global SST class4 bias compared to drifting buoy observations **

Pg. 8, L19: The Fiedler reference is only "in prep". I recommend providing a more thorough explanation with a "personal communication" to Fiedler.

\*\*We propose to add:"The SST spatial variability for each of the experiments was determined by wavenumber auto spectra analysis. All spectra results were calculated on the native grid of the analysis (ORCA025). For each region of interest, the spectra were calculated along the horizontal coordinates closest to a given. The power spectral density of each latitude was calculated using the Welch's average periodogram method, with 28 degrees of freedom (DOF) and Hanning window.The overlapping length for each band averaging was half the band length (Fiedler et al, submitted, personal communication)"\*\*

Pg. 9, Fig. 2d: What is the global mean value of this? It would be helpful to include this on all four panels. It appears to me that it is positive, which seems in conflict with Fig. 3b that shows CPLDA has a more negative global near-surface temperature increment.

\*\*Global mean values have been added to all of the panels (at present to the legend but will be added to the figure titles before the final manuscript is submitted ) The global mean value of the difference between CPLDA and FOAM surface T increments is -0.01K which is consistent with Fig 3b. \*\*

Pg.10, L5-14: This asymmetrical effect of SST increments is quite an important difference between CPLDA and FOAM and warrants further explanation. Perhaps an example figure showing profiles and how the increments deteriorate the profile. However,the argument is based on the fact that large increments in CPLDA then create a large asymmetrical effect on SST. But why are the increments larger in the first place?

\*\*This section has been re-written, see below. The larger increment is likely trying to correct a warm temperature bias due to the under estimation of wind stress in CPLDA.

[revised manuscript text omitted]

Pg. 10, Fig. 3: What is the explanation for the large increments between 50-100m and associated increase in RMSE in CPLDA? Is it related to the asymmetric effect of SST increments? If so, this should be described.

\*\*This section has been re-written (see above) and the increased RMSE addressed.\*\*

Pg. 10, L16: How is the mixed layer depth calculated? The Fig. 4 caption quotes Kara (2000), but some explanation of the methodology used would be appropriate.

\*\*For the stats presented the Kara mixed layer uses a density based criteria. It takes the top most observed profile density (calculated from T and S using the Jackett and McDougall 1994 equation of state) (as long as it is shallower than 10m ) and calculates the depth of the first observation where the density is (first) greater than the density resulting from an decrease in temperature of 0.8 degrees. C compared to the surface value. A linear interpolation of the depth of this observation and the one above gives

one observation of MLD from one temperature and salinity profile observation. (No observation results from a temperature only profile). The same procedure is done for the model equivalent temperatures and salinities which gives the model equivalent MLD. The results of many profiles are gathered to give the statistics of MLD.**

Pg. 10, L17: "mean error .. is -5.2m". The text (Pg. 10, line2) notes that CPLDA has a deeper mixed layer than FOAM, shouldn't this value be positive?

**The error calculated is observation minus model that why the values are negative. We agree it is not clear and will change the text accordingly (5.2 m too deep instead of -5.2).**

Pg. 11, Fig. 4b: Please state whether positive values are deeper or shallower in CPLDA.

**Positive value mean CPLDA is deeper. We will add this to the legend of the figure.**

Pg. 11, L2-4: It is difficult to follow this reasoning when we haven't yet seen the differences in wind stress. This section would be easier to follow if Sec3.4 was presented first providing context regarding differences surface fluxes.

**We agree with the reviewer and we will change the order of the sections by putting the surface fluxes section before the velocities.**

Pg. 11, L7: "Predictions of ocean current are important for marine activities". It would be helpful to expand somewhat since this statement is vague. Also, it should be "currents".

**We propose to add: "Predictions of ocean currents are important for marine activities. They are used for a number of practical applications such as ship routing, marine search and rescue, pollution monitoring, offshore oil and gas industry, marine renewable energy."**

Pg. 11, L19: Are differences in tropical Pacific along capable of affecting global statistics this much? Or are there large differences elsewhere as well?

\*\*Yes, there is no statistically significant differences in the other regions. We propose to add :

"In other regions than the Tropical Pacific no significant differences are observed, CPDLA and FOAM statistics are similar." \*\*

Pg. 11, L21: If the shorter window affects the increments in a direct manner that influences the results to this degree, I think it warrants some further description. Perhaps some examples of increments in this area. This is extremely relevant to the overall effort to produce a "weakly coupled" data assimilation system and should be elaborated upon.

\*\*It is thought that this problem is less to do with the shorter window and more to do with the availability of SLA observations due to the timeliness of the obs sources assimilated, the updated scheduling somewhat alleviates the problem. This will be confirmed by the assimilation window experiments detailed above. \*\*

Pg. 11, L21: By how much are the number of observations limited? This too could be expanded upon to clarify the impact of the assimilation window on results.

\*\*Plot of the number of SLA observations has been added to the SLA section, there is also a slight reduction in the no of profile observations assimilated in CPLDA (5-10%).\*\*

Pg. 12, L3: If the operational system uses the updated scheduling than why not evaluate it instead of the experiment with the reduced number of observations?

\*\*The schedule of the operational system has been updated as a result of the assessment made with the 2015 experiment as it highlighted than the scheduling at the time limited the number of observations impacting especially the SLA and currents. At the time of writing we only had a short period of the operational system with the updated schedule. \*\*

Pg.12, L7: If the unrealistic currents are not always caused by the reduced number of SLA observations, than perhaps the observations are simply covering up an existing issue in the coupled model. This issue warrants further investigation, it its central to many of the results presented.

**The issue is to do with not having the profile observations to constrain the changes in density structure caused by the assimialtion of SLA observations, it is hoped the experiments looking at the assimilation window length will shed some light on this.**

Pg. 14, L8-9: A "reduced heat loss" should lead to warmer SSTs shoudn't they?

**This is a mistake it should say:" The reduced heat gain compared to FOAM between 30S and 30N contributes to the colder SST observed in CPLDA"**

Pg. 14, L9-10: While the differences in short wave radiation and latent heat clearly contribute to differences in net heat flux difference. These fields both have large scale patterns, whereas there is a notable small scale pattern in the net heat flux difference not explained.

**The difference is due to the difference in the representation of the mesoscale between the 2 models, eddies are not located in the same exact positions as there are not completely constraint by the observations (DA).**

Pg. 17, L25-27: As noted above, the argument about the noisier increments is quite important to the overall conclusions. I would recommend a deeper analysis be made of this issue.

**See above, this is being investigated.**

Pg. 19, L8: should be "well-placed"

**It will be corrected**

[Figure]

**Fig. 1.** SLA class4 statistics with respect to CMEMS satellite product (Jason2, Cryosat and Altika) for CPLDA and FOAM. a. For the long CPLDA experiment in 2015 with the old scheduling. b. from the operational

[Figure]

**Fig. 2.** Number of SLA observations assimilated in FOAM and CPLDA in the 2015 experiment

[Figure]

**Fig. 3.** Time series of Sea Ice extent (left) and sea ice volume (right) for CPLDA (2015 experiment) and FOAM

[Figure]

**Fig. 4.** Sea ice extent forecast for CPLDA (2015 experiment) and FOAM during the melting season (February in Antarctic and August in Arctic)

arch/ivtt/godae/

**Fig. 5.** For information not inclusion

---

## Author Response (AR1)

We are grateful to Reviewer1 for the time and efforts assessing our work, and for providing comments.

*These comments are reproduced below, in italics.*

**Responses and modifications proposed to the manuscript are shown in bold.**

*General comments:*

*This study presents a new ocean-atmosphere coupled operational system (CPLDA) with weakly coupled data assimilation developed at Met Office. In the first part, the model components are described and compared to previous coupled and uncoupled systems (GloSea and FOAM respectively). The weakly coupled data assimilation method from Lea et al. (2013, 2015) is also detailed. This system is operated and evaluated during one year (2015) with 6-hour analysis and daily 7-day forecast. The system performances are then compared with the operational ocean-only FOAM (Ryan et al. 2015) and Mercator PSY4 systems and with different observational (SST, MLD, 15 m velocities, SLA) datasets. Atmospheric forcings and turbulent fluxes are also analysed. In summary, the CPLDA system performs as well as ocean-only systems despite its increased complexity and constitute a promising first step toward a fully coupled operational system.*

*The manuscript provides a precise description of the CPLDA system, underlining the benefits and limitations compared to other MetOffice systems and observations. It is globally well written and scientific analyses are seriously presented. However, I recommend the following modifications to improve the manuscript quality and understanding. I have the feeling that some sensitivity tests and analysis are still missing to better cover the system description and validation. This is mainly due to the fact that besides the ocean model similar configurations between CPLDA and FOAM, there a large number of differences between the two systems. Consequently, it is sometimes difficult for the authors to assess precisely the differences between the systems because too much parameters change at the same time. Some additional sensitivity tests allow to better understand those differences (for example, the one month experiment is a different scheduling for the DA). But others sensitivity tests are missing to clearly understand and to disentangle the numerous modifications between the systems. As a consequence, some analysis are not very convincing because of this lack of sensitivity experiments. Depending on the authors will and capacity to run these additional simulations, some corrections will have to be made to the text to confirm or infirm the hypothesis proposed. If no supplementary experiments can be done, some comments must be added to the text to emphasize the limitations of the study. Because atmospheric forcings and surface fluxes are strongly related to SST, upper temperature and MLD, the section 3.4 should be moved before the section 3.3 about velocities (that I would rename "upper or ML velocities"). It is not always clear if the authors are talking about the analysis, the forecast or both in the whole manuscript. It makes the study understanding less clear to follow and more difficult to understand. I think the manuscript can be easily improved by clearly stating this.*

**Experiments have been undertaken using the ocean-only FOAM system to look at the impact of reducing the assimilation window from 24 to 6 hours. These experiments have helped to decompose the effect of the shortened assimilation window from the effect of the coupling and aid in the interpretation of the comparisons to the FOAM system in the manuscript. It is important to note that the scheduling of the operational CPLDA system and the availability of**

observations will not be investigated in these experiments. These experiments have been described in the manuscript and referred to in the relevant sections

We agree with reviewer1 about the organisation of the paper and we will move the section on surface fluxes before the velocities.

*Specific comments:*
*p.1 - Introduction: The introduction is not really relevant because it is mostly a repeat of the system description in section 2. The introduction should be improved to better describe the context of this study, i.e to describe the main physical and technical arguments in favour of developing a coupled system with weakly coupled assimilation (instead of independent systems and assimilations). A description of what is done in other operational centres (ECMWF, NCEP, . . .) could also help to better understand the framework and the interest of the system presented here.*

To provide more context the Met Office efforts, we will update the introduction and in particular we propose to add:

"Coupled systems are used in wide range of applications (short and medium range forecasts, seasonal forecasts, climate prediction and future scenario projections) and improving the initialization of these systems can play a significant part to reduce the development of errors. Using separate atmosphere and ocean analyses to initialize a coupled system can result in an imbalanced system. The imbalance can cause an initialisation shock which potentially increases the development of errors during the forecast. Using a coupled data assimilation (DA) approach has been shown to reduce this initialisation shock (Mulholland et al., 2015). "

and

"In recent years, several centres have developed operational coupled data assimilation systems. Penny and Hamill (2017) provide an overview of many of these efforts, showing the diversity of approaches available. For example, the European Centre for Medium-Range Weather Forecast (ECMWF) has developed a weakly coupled ocean-atmosphere data assimilation system (Browne et al., 2019) which is now operational. This system is shown to reduce forecast errors compared with forecasts initialised from uncoupled analyses. The National Centers for Environmental Prediction (NCEP) is using a coupled data assimilation system for seasonal and sub-seasonal predictions (Saha et al., 2014). At JAMSTEC, a low resolution strongly coupled system was developed to be used for experimental seasonal and decadal prediction, while the US Naval Research Laboratory coupled model is initialised from separate analyses but uses a high-resolution ocean component (1/25). Similarly, the Canadian Centre for Meteorological and Environmental Prediction operational global coupled system uses uncoupled analyses but taking great care to maintain consistency of surface boundary conditions between components in order to avoid initialisation shocks. Smith et al. (2018) show good forecast performance

**compared to the atmosphere-only counterpart of this system, with the most significant impact associated with decreased tropical cyclone intensification.**

**"**

*p.3 l.10-12: it is not clear if VarBC is activated or not in the present study, please specify this in the text.*
**We propose to add: "all the CPLDA experiments in this paper use VarBC."**
*p.5 l.30: please justify why such changes regarding IAU and observation operator were made in CPLDA compared to FOAM.*
**3-hour IAU are used so for the 00Z analyses the full increments will have been added to the background. These analyses can subsequently be used by groups running initialised coupled forecasts.  We propose to add:**

**"This choice is in part to ensure that on the 00Z cycle (with window from 21Z to 03Z) the full increments have been added to the ocean state by 00Z. Hence these ocean restarts can subsequently be used, in exactly the same way as 00Z ocean restarts from FOAM, by groups running initialised coupled forecasts."**

*p.6 l.20: scheduling of the operational system ?*
**it will be corrected**

*p.6 l.25: I don't understand why some results should come from the operational version of the system and others from "test" version ? I find this confusing. Perhaps it should be better to talk about the actual operational version only in the conclusion to avoid any confusion.*
**We propose to include a paragraph on the SLA to the paper. The results from the test experiment 2015 highlighted the degradation of the SLA statistics compared to FOAM cause by the scheduling. The results from the operational system with updated scheduling are presented only to highlight how this problem was solved.**

*p.6 l.31: please add Lellouche et. al 2018 as a reference for Mercator-Ocean PSY4 system.*

**This reference will be added :**
**Lellouche, Jean-Michel & Greiner, Eric & Le Galloudec, Olivier & Garric, Gilles & Regnier, Charly & Drevillon, Marie & Mounir, Benkiran & Testut, Charles-Emmanuel & Bourdalle-Badie, Romain & Gasparin, Florent & Hernandez, Olga & Levier, Bruno & Drillet, Yann & Remy, Elisabeth & Traon, Pierre-Yves. (2018). Recent updates to the Copernicus Marine Service global ocean monitoring and forecasting real-time 1/12° high-resolution system. Ocean Science. 14. 1093-1126. 10.5194/os-14-1093-2018.**

*p7 l.9-11: I think these results are important and should be added to the paper as a distinct section or paragraph with a dedicated figure to illustrate the differences in terms of SLA bias and RMSE between FOAM and the 2 versions of CPLDA with the*

*different schedulings. It is also unclear if the results presented here are those from the current operational system or from the version used in the study ("old scheduling").*

**We agree with the comment and will add a figure to illustrate these results and make clear that the version used for this assessment uses the old scheduling. We propose to add:**

**"CPLDA 2015 and FOAM sea surface height are assessed against observations using class4 statistics (Ryan et al., 2015). The observations used are provided by CMEMS and include data from Altika, CryoSat-2 and Jason-2 satellites. Altimeter bias correction is applied to the observations. For each model comparison against the satellite observations, the model's own altimeter bias is used. This is important because the altimeter bias contains information from the model mesoscale so correcting observations using the altimeter bias from one model to assess a second model penalises this second model.**

**Fig 1 shows a timeseries of the sea level anomaly (SLA) difference statistics assessed against CMEMS observations. In the 2015 experiment with the original scheduling, CPLDA SLA root-mean-squared error (RMSE) is significantly larger than FOAM SLA RMSE (Fig 1a). The larger RMSE in CPLDA can be attributed to the difference in the number of SLA observations assimilated.**

**Fig 2 shows the number of observations assimilated daily by both systems in 2015; the number assimilated by CPLDA is significantly smaller than the number assimilated by FOAM. Differences in scheduling can explain this the CPLDA 2015 experiment "best analysis" runs earlier in the day than the comparable FOAM analysis so fewer observations are available.**

**Following these results, the scheduling of the CPLDA operational system was updated in April 2018. The best analyses at 06Z, 12Z and 18Z are now delayed allowing more observations to be assimilated, as already described in section 2. This change, along with a change in the Met Office database (MetDB) allowing a more frequent ingestion of SLA observations, has resulted in a significant reduction of the CPLDA RMSE (see Fig 1b). The experiments using the FOAM system to look at the effect of the shorter assimilation window in CPLDA showed no significant difference in SLA statistics between the 6 and 24 hour systems."**

[Figure]

**Figure 1: SLA class4 statistics with respect to CMEMS satellite product (Jason2, Cryosat-2 and Altika) for CPLDA and FOAM. a. For the long CPLDA experiment**

**in 2015 with the old scheduling. b. from the operational systems for 2018, the CPLDA scheduling was updated to allow more observations to be assimilated in April 2018.**

[Figure]

**Figure 2: Daily number of SLA observations assimilated in FOAM operational system for 2018 (black line, foam_opfc) and in the CPLDA 2015 experiment (blue line, cplda)"**

*p.14 section 3.4 should be moved close to the sections 3.1 and 3.2 as it is directly related to changes in SST and MLD. It would improve the readability of the manuscript.*
**We agree with this comment and will change the order of the sections accordingly.**

*p.7 l.28-30 This hypothesis should be directly tested by doing some additional sensitivity experiments using a 24h window with CPLDA or a 6h window with FOAM to confirm it or not.*
**Experiments carried out using an ocean-only model to look at the impact of reducing the assimilation window from 24 to 6 hours. Manuscript has been updated accordingly.**

*p.7 l.30 "in CPLDA" -> "in CPLDA analysis": it is not clear when the author is talking about the analysis or the forecast. Please specify it everywhere it is necessary in the text.*
**Manuscript updated accordingly**

*p.7 l.31: is there a way to measure this "overfitting" ? It is not clear why FOAM SST forecast performs better than CPLDA and how is it related to this overfitting.*
**Overfitting can be measured by withholding a sub-set of observations from the DA, these are then used for validation. This approach is not pragmatic in an operational setting where you want to get the best analysis possible at runtime. The magnitude of this overfitting will be assessed in future experiments. These experiments will include investigating scaling the background errors for the 6-hour window (current values calculated from system using 24 hour window). It is thought that in the current CPLDA system background errors are too large (less weight in the DA) thus observations are given too much weight in the assimilation and introducing observation noise.**

**Similar performance is seen in CPLDA and FOAM forecasts at lead times of 36 hours and beyond. The manuscript has been updated to clarify this.**

*p.8 fig.1: the forecast duration is 7 days (168h), please expand the figure axis accordingly*
*p.8 l.19: it could be interesting to give the effective SST resolution for both system even without showing the figure to get an idea of the scale range resolved.*
**The effective SST resolution is the same for both system around 50km. This statement has been added to the manuscript.**

*p.9 section 3.2: please add PSY4 in the analysis and Figure 3 if possible to be more coherent with others sections and to get a more complete comparison between products.*
**Although this would be a useful addition to the manuscript it would require a substantial amount of data processing. As the scope of the paper is the CPLDA system the comparison to FOAM is valuable as the systems are relatively similar which isn't the case for PSY4.**

*p.9 l.9: please give the reference associated with this product.*
**We have added the reference to ARGO product:**
**INSITU_GLO_NRT_OBSERVATIONS_013_030**

*p.9 l.11: please give the range of the RMSE increase to be able to compare it to the SST RMSE.*
**T rmse increased from 0.67 at day1 to 0.73 at day5. This has been added to the manuscript.**

*p.10 l.15: the warm bias and associated significant RMSE (_1_C) located at _100m is not discussed nor explained in the text. Please add a paragraph about this. If the discussion about King et al. 2018 results explain it, it should appear more explicitly in the text.*
**This section has been re-written (see below) and the reviewer comments on RMSE addressed.**

**"Temperature and Mixed Layer Depth**

[revised manuscript text omitted]

*p.10 l.17: are you talking about analysis or forecast ? Please detail both aspects regarding MLD.*
**We have updated the text to make it clear that the MLD results presented are from the analysis. During the forecast the rmse and bias persist (not shown).**

*p.11 l.5: I totally agree with the author: an additional experiment using FOAM with a 6h window or CPLDA with a 12h window is needed to disentangle this effect from other possible factors such as atmospheric forcings, turbulent fluxes schemes, . . . Hence, I strongly suggest to the author to do this simulation if the modifications of the systems are not too heavy and depending on available computing resources. I would greatly improve the manuscript discussion and strengthen the results presented here.*
**These experiments have been carried out, see 1st comment.**

*p.11 section 3.3: please add a word if the product is also assimilated or other products related to oceanic currents. Please make a better distinction between analysis and forecast biases and RMSE.*
**The currents are not assimilated. We have updated the text to make clear what is analysis what is forecast**

*p.11 l.20: the sentence is in contradiction with what is stated above at p.11 l.15-16 ("bias and RMSE stable during forecast"). Please correct this or give more explanations.*
**The bias and RMSE are stable during the forecast on a global scale but in the tropical pacific where there are large erroneous currents in the analysis the RMSE decreases during the forecast.**

*p. 12 l.1-2: is it possible to confirm this statement by comparing directly the number of assimilated observations between both systems please ? Would it be possible to conduct the same kind of tests to address other questions or comments in the manuscript related to the difference in term of assimilation time window or observation number ?*
**In the new paragraph on SLA we will add a plot highlighting the difference in the numbers of observation between FOAM and CPLDA 2015 experiments. The question of the assimilation time window has been investigated by running an experiment with FOAM using a 6-hour window, the text has been updated with these results. i.e.,**
**"The experiments using the FOAM system to look at the effect of the shorter assimilation window in CPLDA showed no significant difference in SLA statistics between the 6 and 24 hour systems."**

*p.12 l.6-7: again, an additional test with a different time window would strongly clarify these statements and give a real matter for discussion.*
**Yes as above**

*p.12 l.10-11: again, an additional 1-month experiment using the new MDT would clarify this.*

**The MDT will not be updated in the current iteration of the CPLDA system so these experiments will not be undertaken. The next generation O-A coupled system which is being developed currently uses an updated MDT from CNES-CLS09 to CNES-CLS13 and initial results look promising but these results are outside the scope of this paper. The final implementation will likely use CNES-CLS18 but this is yet to be tested.**

*p.12 fig. 5:Figure 5 is not described nor employed in the text. Please supress it or comment it.*
**We will remove Figure 5**

*p.13 l.2-3: on the contrary, it has been shown that coupled models have more EKE damping than ocean-only forced models (see Renault et al. 2016 for example). Consequently, this explanation is incomplete or erroneous.*
**We agree with the reviewer; the text is not clear. We propose to update it:**

**"Eddy Kinetic Energy (EKE) was also compared in CPLDA, FOAM and PSY4. In an ocean-only model, using the ocean velocity to calculate the wind stress has a damping effect on the eddies (Duhaut and Straub, 2006; Dawe and Thompson, 2006; Renault et al., 2016). Despite the expected reduced damping effect in a coupled system, we did not observe a higher EKE in CPLDA than in FOAM. However at ¼ deg resolution, the mesoscale is poorly represented and most of the EKE is injected into the model by the SLA assimilation. Hence differences in EKE between CPLDA and FOAM are mainly caused by differences in SLA observations assimilated, or impacts of the shorter assimilation window, rather than due to a reduced eddy damping by the wind stress."**

*p.14-15 – section 3.4: There is no numbers in this section text to quantify the differences between atmospheric forcings heat fluxes and stresses. It will make the comparison easier and more "physical" if you add them.*
**We calculated the average value for the flux differences plotted in Figure 8. On average, CPLDA loses more heat than FOAM which contributes to the colder SST observed in CPLDA. The differences in short wave radiation show some strong patterns with the largest difference in the eastern boundary upwelling but on average its contribution is limited. On average for the global ocean, the strongest contributions come from the latent heat differences.**

**Average for total heat flux diff (CPLDA-FOAM) = -3.9 W/m2**
**Average for latent heat diff (CPLDA – CPLDA with CORE)=+9.61 W/m2          (1)**
**Average for latent heat diff (CPLDA – NWP)=-14.50 W/m2                      (2)**
**Average for short wave diff (CPLDA-FOAM) =+1.05 W/m2                        (3)**
**(1)+(2)+(3) = -3.84 W/m2**

**These values have been added to the Figure 8 captions and are also referred to in the text.**

*p.15 l.1-3: the shortwave (SW) bias pattern suggest an SW overestimation on all eastern boundary upwelling systems. This is usually related to an underestimation of the low-level stratiform clouds which atmospheric models have difficulties to represent. A comment about this atmospheric bias should be added if it is also present in NWP.*

**The stratocumulus clouds are under estimated in Met office atmosphere model and causes a short-wave overestimation on the eastern boundary upwelling systems. The bias gradually reduces as the resolution increases; the resolution is lower in the atmosphere component of CPLDA (40km) than in the NWP fluxes forcing FOAM (17km) and so the overestimation is larger in CPLDA.**

*p.15 l.21: please give a quantification of the contributions of atmospheric state and latent heat to the total net heat flux difference.*

**Globally, the differences in latent heat due to the different bulk formulae (+9.61 W/m2) and the different atmosphere (-14.50 W/m2) are the main contributions to the difference in heat fluxes.**

*p. 15 – wind stress analysis: I have some difficulties to correctly understand this part. First, there is no distinction between atmospheric analysis and forecasts. I suppose analysis are relatively close to MetOp observations as they are probably assimilated, while 7-days forecasts must have quickly increasing surface stress errors. Consequently, it is difficult to guess where the stress differences between CPLDA and MetOp come from. Then, CPLDA wind stress is strongly and globally underestimated (_-0.1 N/m2) compared to MetOp. This bias is far larger than the difference between CPLDA and FOAM (_+0.5 N/m2). I don't think this is related to bulk formulae differences (COARE 3.0 is nearly identical to COARE 3.5 except in very strong wind conditions which statistically almost never occur). It can be partially related to the fact that scaterometers usually provide surface stress in atmospheric neutral stability condition, but it cannot explain the global underestimation observed here. This can also be related to the absolute or relative wind/current coupling in CPLDA. But finally, I don't understand why this global stress underestimation is not associated with warm SST and shallow MLD biases in both FOAM and CPLDA as it should be the case. Please explain clearly why it is not the case.*

**All wind stress presented are from the analysis.**

**Investigations are ongoing as to why the differences to observations are larger than expected. As we are unable to explain the observed differences at present the comparison to MetOp observations has been removed from the figure and the manuscript.**

**The large negative increment suggests the model is too warm, the MLD is constrained by the assimilation of not only profile observations but also the SST observations as described in section 3.2. Without a free-run it's hard to know what the MLD would look like due to wind-stress differences only without the effect of the surface increments on the MLD.**

*p.16 – conclusion: Conclusion should be updated accordingly to the modifications done in the manuscript.*

**We have updated the conclusion**

We are grateful to Reviewer2 for the time and efforts assessing our work, and for providing comments.

*These comments are reproduced below, in italics.*

**Responses and modifications proposed to the manuscript are shown in bold.**

*General Comments:*

*This paper presents an evaluation of the ocean component of an atmosphere-ocean coupled prediction system that has been running in real-time at the UK Metoffice and builds upon the work presented by Lea et al. (2015). In particular, this system employs a "weakly coupled" data assimilation approach, which affects the analysis and forecast quality in a number of ways. A systematic approach is used to provide an objective evaluation of the different state variables (sea surface temperature, mixed layer depth, currents) against accepted reference datasets, including other operational forecast products. The presentation is generally clear, but remains fairly descriptive, limiting the conclusions that can be drawn from the study. Many of the differences between the coupled and operational forecasting systems are due to differences in particular parameter choices and not directly due to the coupled model or "weakly coupled data assimilation" themselves. As a result, the value of the paper is limited to a presentation of the baseline skill of the system. Even this is put somewhat in question, as the results (for currents and sea level) are shown to be sensitive to differences in data assimilated in the two systems resulting in much larger errors in the coupled system.*

 *The paper would benefit significantly from having additional experiments to clarify the impact of the different parameter choices and a more in-depth analysis of results.*

*Additionally, the paper would benefit from some reorganization as the differences in fluxes are described near the end of the paper, much of which is needed to understand the differences being presented in previous sections.*

*I recommend the paper for publication but strongly encourage the authors to deepen the analysis of the results and provide additional experiments to shed-light on the source of differences.*

**Idem reviewer1**

**Experiments have been undertaken using the ocean-only FOAM system to look at the impact of reducing the assimilation window from 24 to 6 hours. These experiments have helped decompose the effect of the shortened assimilation window from the effect of the coupling and aid in the interpretation of the comparisons to the FOAM system in the manuscript. It is important to note that the scheduling of the operational CPLDA system and the availability of observations will not be investigated in these experiments. These experiments have been described in the manuscript and referred to in the relevant sections.**

**We agree with reviewer2 about the organisation of the paper and we will move the section on surface fluxes before the velocities. We will also add a paragraph on the SLA results.**

*Specific Comments:*

*Section 1: The introduction is limited to activities at the UK Metoffice and would benefit greatly from some context into why weakly coupled data assimilation is of interest and previous efforts. For example, a study has recently been published that should be*

*referenced in this discussion (Browne et al., 2019). Moreover, several centers are now running coupled forecasting systems operationally, which should be mentioned to provide context into UK Metoffice efforts. Browne, P.A., de Rosnay, P., Zuo, H., Bennett, A. and Dawson, A., 2019. Weakly Coupled Ocean–Atmosphere Data Assimilation in the ECMWF NWP System. Remote Sensing, 11(3), p.234.*

**To provide more context the Met Office efforts, we will update the introduction and in particular we propose to add:**

**"Coupled systems are used in wide range of applications (short and medium range forecasts, seasonal forecasts, climate prediction and future scenario projections) and improving the initialization of these systems can play a significant part to reduce the development of errors. Using separate atmosphere and ocean analyses to initialize a coupled system can result in an imbalanced system. The imbalance can cause an initialisation shock which potentially increases the development of errors during the forecast. Using a coupled data assimilation (DA) approach has been shown to reduce this initialisation shock (Mulholland et al., 2015). "**

**and**

**"In recent years, several centres have developed operational coupled data assimilation systems. Penny and Hamill (2017) provide an overview of many of these efforts, showing the diversity of approaches available. For example, the European Centre for Medium-Range Weather Forecast (ECMWF) has developed a weakly coupled ocean-atmosphere data assimilation system (Browne et al., 2019) which is now operational. This system is shown to reduce forecast errors compared with forecasts initialised from uncoupled analyses. The National Centers for Environmental Prediction (NCEP) is using a coupled data assimilation system for seasonal and sub-seasonal predictions (Saha et al., 2014). At JAMSTEC, a low resolution strongly coupled system was developed to be used for experimental seasonal and decadal prediction, while the US Naval Research Laboratory coupled model is initialised from separate analyses but uses a high-resolution ocean component (1/25). Similarly, the Canadian Centre for Meteorological and Environmental Prediction operational global coupled system uses uncoupled analyses but taking great care to maintain consistency of surface boundary conditions between components in order to avoid initialisation shocks. Smith et al. (2018) show good forecast performance compared to the atmosphere-only counterpart of this system, with the most significant impact associated with decreased tropical cyclone intensification.**

**"**

*Pg1, L19: Johns et al. (2012) is a technical report that appears not to be publically available. There are a number of such reports referenced here. If these reports are only available upon request, than greatly detail regarding their content should be included in the text. Also for Lea et al. (2013).*

**We propose to add details for both references:**

**"In Johns et al. (2012), a large set of global coupled atmosphere-ocean-sea ice 15-day hindcasts were initialised from separate atmosphere and ocean-ice analyses. Compared to uncoupled forecasts, the coupled 15-day hindcasts show improved forecasting skill especially in the Tropics, a strongly coupled region where small changes in sea surface temperature (SST) can exert a major influence on patterns of convection and remote responses to diabatic heating. Both atmosphere and ocean biases benchmarked against control simulations suggest a more vigorous water cycle, improved large-scale circulation, enhanced convection, stronger teleconnections and somewhat improved representation of the Madden-Julian oscillations. They conclude that significant benefits should arise from full integration of a coupled NWP system."**

**and**

**"The approach taken to develop the CPLDA system is described in detail in Lea et al. (2013, 2015). It is based on the existing coupled model developed for seasonal forecasting and climate prediction and on existing data assimilation systems for the ocean, sea-ice, land and atmosphere. The data assimilation follows a "weakly coupled" approach. This means the coupled model provides background information for separate analyses in each sub-component with the increments being added back into the coupled model. Lea et al. (2013) assessed the performance of the system against uncoupled control experiments for two short trials (December 2011 and June 2012)."**

*Pg2, L14: "To avoid duplication of previous work . . .we did not investigate the impact on the diurnal cycle already covered in Lea et al., (2013)". This is a fairly important point that is referred to later in terms of errors in temperature and mixed layer depth. Since Lea et al., (2013) is only a technical report these results could be included here (or at least explained more clearly).*

**We propose to add:**

**"They investigated the impact of the initialisation on the diurnal cycle of SST by comparing to geostationary satellite data from SEVIRI. They looked at the diurnal SST range and showed that the initialisation of the coupled forecast has very little impact on this. Lea et al. (2015) showed that in the South Pacific the coupled model diurnal cycle errors against observations are somewhat amplified compared to the ocean-only control. However the diurnal cycle is perhaps under-estimated in the central North Atlantic (Lea et al., 2013)."**

*Pg. 3, L1: COARE3.0, please add a reference. Also this is not noted in Table 1.*

**This reference will be added**
**Fairall, C.W., E.F. Bradley, J.E. Hare, A.A. Grachev, and J.B. Edson, 2003: Bulk Parameterization of Air–Sea Fluxes: Updates and Verification for the COARE Algorithm. J. Climate, 16, 571–591, https://doi.org/10.1175/1520-0442(2003)016<0571:BPOASF>2.0.CO;2**

**We have updated table 1 accordingly.**

*Pg. 3, L9: Lea et al. (2015) note significant difficulties associated with runoff. How is runoff handled here? Are there differences between FOAM and CPLDA?*

**Due to the issues in Lea et al. (2015), we are for the moment using the same climatological run off as in FOAM. Since Lea et al. (2015), runoff from the atmosphere/land component may well have improved and we plan to revisit using this in future.**

*Pg. 3, L32: "Haney retroaction". Please provide a reference and more details. I.e. what is the timescale of the relaxation? To what fields?*

**A Haney flux correction (Haney, 1971) is applied to the sea surface salinity (SSS) based on the difference between the model and climatology. The magnitude of the restoring on SSS is -33.33 mm/day/psu. This has been added in the text and in table 1. This reference has also been added:**

**Haney,R. 1971. Surface boundary conditions for ocean general circulation models. J. Phys. Oceanogr. 1: 241–248.**

*Pg.4, L19: It says here that CPLDA uses a fixed freezing temperature, which differs from what is indicated in Table1.*
**Reviewer is correct; table 1 has been corrected.**

*Pg. 5, L27-32: It may be useful to note here that there is also an impact on the observation cutoff time that affects the number of SLA observations used.*
**We propose to add this:**
**"It is also important to note that differences in observation cutoff due to difference in operational scheduling impact the number of observations assimilated by each system."**

*Pg. 6, Sec2.6: If the CPLDA system has been running operationally since October 2016, why is it necessary to run additional experiments? If the aim here is to benchmark the skill of CPLDA as compared to FOAM why not use the operational runs? If iit possible to make additional runs, could a longer run with the SLA observation cutoff corrected be produced? This would make the evaluation much more straightforward.*

**Since October 2016, the operational CPLDA suite had several updates and bug fixes so we could not have a long enough period with consistent settings to be able to properly assess the system. We will not be able to produce a longer run with the corrected cutoff but results from the Operational CPLDA system with the updated scheduling allowing more SLA observations to be assimilated now shows similar SLA statistics than the FOAM system.**

*Pg. 6, L17: "There are a few differences. . .". What is the impact of these differences? If they are not too significant an analysis of the operational run compared to FOAM for Oct. 2016 onwards could add additional insight to the paper.*

There are differences between the operational system in 2016 and the same system in 2018 (several upgrades and bug fixes). The 2015 experiment was made to assess the 2018 version of the operational system. As noted in the manuscript some differences are unavoidable (different observations available) and the other main difference is the scheduling that was update in the operational system in June 2018 as a result of the assessment of the 2015 experiment.

*Pg. 7, L7: SLA is an important variable providing information regarding the circulation and mesoscale activity. It is also relevant here to provide context to the errors in 15m velocity. I would strongly encourage you to include these results, despite the poor results. Ideally, an additional run could be made with the cutoff issue corrected.*

The current CPLDA operational system with the updated scheduling allowing more observations to be assimilated has SLA statistics similar to FOAM. A section on the SLA results will be added to the manuscript.

"CPLDA 2015 and FOAM sea surface height are assessed against observations using class4 statistics (Ryan et al., 2015). The observations used are provided by CMEMS and include data from Altika, CryoSat-2 and Jason-2 satellites. Altimeter bias correction is applied to the observations. For each model comparison against the satellite observations, the model's own altimeter bias is used. This is important because the altimeter bias contains information from the model mesoscale so correcting observations using the altimeter bias from one model to assess a second model penalises this second model.

Fig 1 shows a timeseries of the sea level anomaly (SLA) difference statistics assessed against CMEMS observations. In the 2015 experiment with the original scheduling, CPLDA SLA root-mean-squared error (RMSE) is significantly larger than FOAM SLA RMSE (Fig 1a). The larger RMSE in CPLDA can be attributed to the difference in the number of SLA observations assimilated.

Fig 2 shows the number of observations assimilated daily by both systems in 2015; the number assimilated by CPLDA is significantly smaller than the number assimilated by FOAM. Differences in scheduling can explain this the CPLDA 2015 experiment "best analysis" runs earlier in the day than the comparable FOAM analysis so fewer observations are available.

Following these results, the scheduling of the CPLDA operational system was updated in April 2018. The best analyses at 06Z, 12Z and 18Z are now delayed allowing more observations to be assimilated, as already described in section 2. This change, along with a change in the Met Office database (MetDB) allowing a more frequent ingestion of SLA observations, has resulted in a significant reduction of the CPLDA RMSE (see Fig 1b). The experiments using the FOAM system to look at the effect of the shorter assimilation window in CPLDA showed no significant difference in SLA statistics between the 6 and 24 hour systems."

[Figure]

**Figure 1: SLA class4 statistics with respect to CMEMS satellite product (Jason2, Cryosat-2 and Altika) for CPLDA and FOAM. a. For the long CPLDA experiment in 2015 with the old scheduling. b. from the operational systems for 2018, the CPLDA scheduling was updated to allow more observations to be assimilated in April 2018.**

[Figure]

**Figure 2: Daily number of SLA observations assimilated in FOAM operational system for 2018 (black line, foam_opfc) and in the CPLDA 2015 experiment (blue line, cplda)"**

*Pg. 7, L16: It is argued that sea ice is not evaluated because of a difference in freezing temperature. However, there are differences in many other aspects as well (bulk formulae, SLA observation cutoff, assimilation window). If the aim is to benchmark the performance of the system, than an evaluation of the sea ice component should be included as well.*
**We propose include a figure illustrating the sea ice results.**

**"Sea ice:**

**Fig 3 shows the sea ice extent and volume respectively for CPLDA and FOAM best analyses. It shows that the sea ice extent is similar between CPLDA and FOAM and is also comparable to OSTIA. Fig 3 shows that the sea ice volumes are similar in the Arctic but in the Antarctic CPLDA has a reduced volume compared to FOAM in the Southern Hemisphere winter and spring. Differences in volume can be attributed to differences in sea ice thickness probably caused by differences in freezing temperature treatment. Fig 4 shows the sea**

ice extent forecast for the melting season in Arctic (August) and Antarctic (February) for CPLDA and FOAM. Both models have the tendency to melt too much ice during the forecast especially in the Arctic; in the Antarctic the exaggerated melting in CPLDA is slightly reduced compared to that observed in FOAM. Differences in freezing temperature treatment referred to in 2.3 may have impacted the sea ice simulation."

[Figure]

**Figure 3: Time series of sea ice extent (left) and sea ice volume for CPLDA (2015 experiment) and FOAM**

[Figure]

**Figure 4: Sea ice extent forecast for CPLDA (2015 experiment) and FOAM during the melting season (February in Antarctic and August in Arctic)**

*Pg. 7, L20: Is it the instantaneous SST or a daily average?*
**We use the daily average - it has been clarified in the text.**

*Pg. 8, L9: Should this be 0.01K? The global mean difference in bias between CPLDA and PSY4 appears to be less than 0.05K. Also, if OSTIA is cooler than CPLDA and PSY4 assimilates OSTIA, why is PSY4 warmer than CPLDA?*
**Global average difference between CPLDA and OSTIA is 0.06K; the text has been updated.**

**Using Class4 statistics from the GODEA model inter-comparison (see figure 3) PSY4 (purple) is generally biased warm and OSTIA (grey) is generally biased cold relative to surface drifters, this is consistent with the results presented for 2015. This reasons for the difference in SST characteristics between the systems despite PSY4 assimilating OSTIA could be due to model biases, atmospheric forcing etc but diagnosing these is outside the scope of this paper.**

[Figure]

**Figure 5.  Global SST class4 bias compared to drifting buoy observations**

*Pg. 8, L19: The Fiedler reference is only "in prep". I recommend providing a more thorough explanation with a "personal communication" to Fiedler.*

**We propose to add:**

**"The SST spatial variability for each of the experiments was determined by wavenumber auto spectra analysis. All spectra results were calculated on the native grid of the analysis (ORCA025). For each region of interest, the spectra were calculated along the horizontal coordinates closest to a given latitude. The power spectral density of each latitude was calculated using the Welch's average periodogram method, with 28 degrees of freedom and Hanning window. The overlapping length for each band averaging was half the band length (Fiedler, 2018). The results for CPLDA and FOAM are very similar; in both systems the effective SST resolution is around 50 km."**

*Pg. 9, Fig. 2d: What is the global mean value of this? It would be helpful to include this on all four panels. It appears to me that it is positive, which seems in conflict with Fig. 3b that shows CPLDA has a more negative global near-surface temperature increment.*

**Global mean values have been added to the figure titles.  The global mean value of the difference between CPLDA and FOAM surface T increments is -0.01K which is consistent with Fig 7b (previously Fig 3b).**

*Pg.10, L5-14: This asymmetrical effect of SST increments is quite an important difference between CPLDA and FOAM and warrants further explanation. Perhaps an*

*example figure showing profiles and how the increments deteriorate the profile. However,the argument is based on the fact that large increments in CPLDA then create a large asymmetrical effect on SST. But why are the increments larger in the first place?*

**This section has been re-written, see below. Experiments have shown that the larger increment is likely a symptom of the shorter assimilation window in CPLDA compared to FOAM.**

**"Temperature and Mixed Layer Depth**

[revised manuscript text omitted]

*Pg. 10, Fig. 3: What is the explanation for the large increments between 50-100m and associated increase in RMSE in CPLDA? Is it related to the asymmetric effect of SST increments? If so, this should be described.*
This section has been re-written (see above) and the increased RMSE addressed.

*Pg. 10, L16: How is the mixed layer depth calculated? The Fig. 4 caption quotes Kara (2000), but some explanation of the methodology used would be appropriate.*

**For the stats presented the Kara mixed layer uses a density based criteria. It takes the top most observed profile density (calculated from T and S using the Jackett and McDougall 1994 equation of state) (as long as it is shallower than 10m ) and calculates the depth of the first observation where the density is (first) greater than the density resulting from an decrease in temperature of 0.8 degrees. C compared to the surface value. A linear interpolation of the depth of this observation and the one above gives one observation of MLD from one temperature and salinity profile observation. (No observation results from a temperature only profile). The same procedure is done for the model equivalent temperatures and salinities which gives the model equivalent MLD. The results of many profiles are gathered to give the statistics of MLD.**

*Pg. 10, L17: "mean error .. is -5.2m". The text (Pg. 10, line2) notes that CPLDA has a deeper mixed layer than FOAM, shouldn't this value be positive?*

**The error calculated is observation minus model that why the values are negative. We agree it is not clear and have changed the text accordingly (5.2 m too deep instead of -5.2).**

*Pg. 11, Fig. 4b: Please state whether positive values are deeper or shallower in CPLDA.*

**Positive value mean CPLDA is deeper. We have added this to the legend of the figure.**

*Pg. 11, L2-4: It is difficult to follow this reasoning when we haven't yet seen the differences in wind stress. This section would be easier to follow if Sec3.4 was presented first providing context regarding differences surface fluxes.*

**We agree with the reviewer and have changed the order of the sections by putting the surface fluxes section before the velocities.**

*Pg. 11, L7: "Predictions of ocean current are important for marine activities". It would be helpful to expand somewhat since this statement is vague. Also, it should be "currents".*

**We propose to add:**

**"Predictions of ocean currents are important for marine activities. They are used for a number of practical applications such as ship routing, marine search and rescue, pollution monitoring, offshore oil and gas industry, and marine renewable energy."**

*Pg. 11, L19: Are differences in tropical Pacific along capable of affecting global statistics this much? Or are there large differences elsewhere as well?*

**Yes, there is no statistically significant differences in the other regions. We propose to add :**

**"In regions other than the Tropical Pacific no significant differences are observed, with CPDLA and FOAM statistics similar."**

*Pg. 11, L21: If the shorter window affects the increments in a direct manner that influences the results to this degree, I think it warrants some further description.*

*Perhaps some examples of increments in this area. This is extremely relevant to the overall effort to produce a "weakly coupled" data assimilation system and should be elaborated upon.*

**It is thought that this problem is less to do with the shorter window and more to do with the availability of SLA observations due to the timeliness of the obs sources assimilated (the updated scheduling somewhat alleviates the problem). This has been confirmed by the assimilation window experiments detailed above: very little differences were observed in SLA statistics between the 6 and 24 hour assimilation window experiments.**

*Pg. 11, L21: By how much are the number of observations limited? This too could be expanded upon to clarify the impact of the assimilation window on results.*

**Plot of the number of SLA observations has been added to the SLA section, there is also a slight reduction in the no of profile observations assimilated in CPLDA (5-10%).**

*Pg. 12, L3: If the operational system uses the updated scheduling than why not evaluate it instead of the experiment with the reduced number of observations?*

**The schedule of the operational system has been updated as a result of the assessment made with the 2015 experiment as it highlighted than the scheduling at the time limited the number of observations impacting especially the SLA and currents. At the time of writing we only had a short period of the operational system with the updated schedule.**

*Pg.12, L7: If the unrealistic currents are not always caused by the reduced number of SLA observations, than perhaps the observations are simply covering up an existing issue in the coupled model. This issue warrants further investigation, it its central to many of the results presented.*

**The issue is to do with not having the profile observations to constrain the changes in density structure caused by the assimilation of SLA observations. Similar model blow-ups were not observed during the 6 month experiments looking at the assimilation window length.**

*Pg. 14, L8-9: A "reduced heat loss" should lead to warmer SSTs shoudn't they?*
**This is a mistake it should say:" The reduced heat gain compared to FOAM between 30S and 30N contributes to the colder SST observed in CPLDA"**

*Pg. 14, L9-10: While the differences in short wave radiation and latent heat clearly contribute to differences in net heat flux difference. These fields both have large scale patterns, whereas there is a notable small scale pattern in the net heat flux difference not explained.*
**The difference is due to the difference in the representation of the mesoscale between the 2 models, eddies are not located in the same exact positions as there are not completely constraint by the observations (DA). We have added the following text:**

**"The small scale pattern differences are due to differences in the representation of the mesoscale between the two models (eddies are not completely**

**constrained by the observations and so are not located in exactly the same positions)."**

*Pg. 17, L25-27: As noted above, the argument about the noisier increments is quite important to the overall conclusions. I would recommend a deeper analysis be made of this issue.*
**See above, this has been investigated.**

*Pg. 19, L8: should be "well-placed"*

**It has been corrected**

[revised manuscript text omitted]